

# Equilibrium simulations of Marine Isotope Stage 3 climate

Chuncheng Guo[1], Kerim H. Nisancioglu[2,3], Mats Bentsen[1], Ingo Bethke[1], and Zhongshi Zhang[1]

[1]NORCE Norwegian Research Centre, Bjerknes Centre for Climate Research, Bergen, Norway
[2]Department of Earth Science, University of Bergen, and Bjerknes Centre for Climate Research, Bergen, Norway
[3]Centre for Earth Evolution and Dynamics, Department of Geosciences, University of Oslo

*Correspondence to:* Chuncheng Guo (chuncheng.guo@norceresearch.no)

**Abstract.**

An equilibrium simulation of Marine Isotope Stage 3 (MIS3) climate with boundary conditions characteristic of Greenland Interstadial 8 (GI-8; 38 ka BP) is carried out with the Norwegian Earth System Model (NorESM). A computationally efficient configuration of the model enables long integrations at relatively high resolution, with the simulations reaching a quasi-equilibrium state after 2500 years. We assess the characteristics of the simulated large-scale atmosphere and ocean circulation, precipitation, ocean hydrography, sea ice distribution, and internal variability. The simulated MIS3 interstadial near surface air temperature is 2.9°C cooler than the pre-industrial (PI). The Atlantic Meridional Overturning Circulation (AMOC) is deeper and intensified (by $\sim 13\%$). There is a decrease in the volume of Antarctic Bottom Water (AABW) reaching the Atlantic. However, there is an increase in ventilation of the Southern Ocean, associated with a significant expansion of Antarctic sea ice and intensified brine rejection, invigorating ocean convection. In the central Arctic, sea ice is $\sim 2$ m thicker, with an expansion of sea ice in the Nordic Seas during winter. Simulated MIS3 inter-annual variability of the El Niño-Southern Oscillation (ENSO) and the Arctic Oscillation are weaker compared to the pre-industrial. Attempts at triggering a non-linear transition to a cold stadial climate state by varying atmospheric $CO_2$ concentrations and Laurentide Ice Sheet height, suggest that the simulated MIS3 interstadial state in the NorESM is relatively stable, thus questioning the potential for unforced abrupt transitions in Greenland climate during the last glacial.

## 1 Introduction

Marine Isotope Stage 3 (MIS3), a period about 60 – 30 ka BP (thousand years before present) during the last glacial, was characterised by millennial-scale abrupt climate transitions. These events are known as Dansgaard-Oeschger (D-O) events, as revealed by the Greenland oxygen isotope ice core records (Dansgaard et al., 1993). A D-O event consists of an abrupt transition from a cold stadial climate state to a relatively warm interstadial climate state, followed by a gradual return to cold stadial conditions (Huber et al., 2006). Correlated with the rapid change of Greenland temperature, warming of up to 15 °C within a few decades during the stadial-to-interstadial transition, the North Atlantic and Nordic Seas are subject to abrupt climate transitions as interpreted from a number of marine sediment cores (Bond et al., 1997; Rasmussen and Thomsen, 2004; Dokken et al., 2013). Towards the end of every few stadial periods, the marine sediments show evidence of massive calving of the Laurentide Ice Sheet, with large numbers of icebergs transversing and melting in the North Atlantic. These events



are known as Heinrich events (Heinrich, 1988). The freshwater from these melting icebergs is thought to have weakened the Atlantic Meridional Overturning Circulation (AMOC), possibly causing further cooling of the Northern Hemisphere (e.g. Broecker, 1994).

While there has been significant advances in our understanding of the dynamics behind D-O events in recent years, the
key mechanisms triggering these abrupt climate transitions are still under debate and remain illusive. A leading hypothesis is related to a switch between strong, weak, and off modes of the AMOC (Rahmstorf, 2002; Böhm et al., 2015; Henry et al., 2016). This has the potential to significantly alter the circulation and northward heat transport in the Atlantic Ocean. Model based studies have shown that changes in the mode of the AMOC can be triggered by, e.g., freshwater input from melting ice sheets (Ganopolski and Rahmstorf, 2001), variations in the size of the Laurentide Ice Sheet (Zhang et al., 2014), as well
as changes in atmospheric $CO_2$ (Klockmann et al., 2018). Another theory for explaining the abrupt warming of Greenland invokes atmospheric circulation changes triggered by transitions in sea ice cover: e.g., Li et al. (2005, 2010) showed that shifts in Greenland precipitation and temperature are consistent with the climate response induced by sea ice growth and retreat, in particular over the Nordic Seas. The sea ice acts as a lid, insulating the ocean from the atmosphere and reducing the amount of heat released. Proxy data from sediment cores in the Nordic Seas (Rasmussen and Thomsen, 2004; Dokken et al., 2013;
Ezat et al., 2014) suggest that the warm Atlantic inflow can be separated from the sea surface by a halocline, and slowly accumulate heat in the subsurface and intermediate/deep waters during MIS3 stadials. Eventually the warming below the halocline destabilises the water column, and brings warm Atlantic water to the surface, tipping the Nordic Seas into an ice-free state that can lead to a rapid warming as seen in the Greenland ice cores (e.g., Dokken et al., 2013).

Although prescribed external forcing are often introduced into the model to trigger D-O events (e.g., Ganopolski and Rahm-
storf, 2001; Menviel et al., 2014), several model simulations in recent years have been reported to be able to spontaneously reproduce rapid cold-to-warm transitions that resemble D-O events. The underlying mechanisms for these self-sustained oscillations in the models involve, for instance, the "kicked" salt oscillator acting in the Atlantic (Peltier and Vettoretti, 2014) and a similar type of "thermohaline oscillator" (Brown and Galbraith, 2016), stochastic atmospheric forcing (Kleppin et al., 2015), and the formation of North Atlantic super polynyas and subsequent heat release (Vettoretti and Peltier, 2016).

Most studies of D-O events apply a coupled model configured with last glacial maximum (LGM), or pre-industrial (PI) boundary conditions. Very few model studies apply MIS3 boundary conditions. These MIS3 studies include models with varying resolution and complexities, including an atmospheric general circulation model forced with fixed SSTs (Barron and Pollard, 2002), a coupled model of intermediate complexity (Van Meerbeeck et al., 2009), and a higher-resolution atmospheric and oceanic general circulation model (Brandefelt et al., 2011). Van Meerbeeck et al. (2009) include a stadial and an interstadial
simulation and study the model response to changes in orbital forcing and greenhouse gases (GHG), typical of MIS3 conditions; they conclude that neither orbital nor GHG forcing can explain the D-O like variability. Rather, a freshwater input to the surface ocean must be invoked to move the system from an (equilibrium) interstadial state to a (perturbed) stadial state. While a strong AMOC was simulated by Van Meerbeeck et al. (2009), Brandefelt et al. (2011) found an AMOC slowdown of approximately 50% in their MIS3 Greenland Stadial 12 (GS-12; 44 ka BP) configuration. The large reduction in AMOC, and relatively cold
North Atlantic SST, is simulated without freshwater forcing. Brandefelt et al. (2011) suggest that the MIS3 equilibrium state



is highly model-dependent. Note however, that a dramatic AMOC weakening during non-Heinrich stadials is not supported by geological reconstructions (e.g., Böhm et al., 2015).

In this work, we present a MIS3 interstadial equilibrium simulation employing a new version of the Norwegian Earth System Model (NorESM) designed for multi-millennial and ensemble studies (Guo et al., 2018). The state-of-the-art Earth system model features a 2° horizontal resolution for the atmosphere and land components, and a 1° horizontal resolution for the ocean and sea ice components. A faster model throughput of approximately 90 model years per day can be achieved with existing hardware, compared to <40 model years per day for the CMIP5 version of NorESM with comparable resolution and hardware. As documented by Guo et al. (2018), this NorESM version shows improved skill compared to the CMIP5 version in simulating AMOC and sea ice thickness/extent, both important quantities when discussing MIS3 climate and the dynamics of D-O events.

We configure the model with boundary conditions characteristic of 38 ka BP, immediately following the onset of Greenland Interstadial 8 (GI-8). Spontaneous occurrence of D-O like abrupt climate transitions are not simulated in the model during the 2500 year integration using MIS3 boundary conditions. Instead, the experiment serves as a baseline simulation for evaluating equilibrium interstadial climate states during MIS3. A satisfactory quasi-equilibrium state is reached before the end of the long integration, and we assess the basic interstadial climate state of the atmosphere, ocean, and sea ice as represented by the model.

With a realistic model configuration using MIS3 boundary conditions, a long equilibrated integration, and a model with relatively high resolution and encouraging physical performance, we aim to improve our understanding of MIS3 climate and provide a baseline for further sensitivity studies exploring the dynamics of D-O events. The structure of the paper is arranged as follows: in Section 2, we give a brief overview of the NorESM, including details of the new version used in this study, followed by a description of the MIS3 experimental configuration; in Section 3, we assess the equilibrium state of the MIS3 long integration with NorESM, followed by details of the simulated mean MIS3 interstadial state of the atmosphere, ocean, sea ice, and internal variability of the model. A parallel experiment with typical boundary conditions of Greenland stadials is presented in Section 4, followed by a discussion on the simulated AMOC and model response to changes in GHG and ice sheet height in Section 5. The main conclusions are summarized in Section 6.

## 2 Methods

### 2.1 The climate model: NorESM

The version of NorESM used in this study is based on the Community Climate System Model version 4 (CCSM4; Gent et al., 2011) but differs from the latter in several aspects (Bentsen et al., 2013): NorESM uses an isopycnic coordinate ocean model that originates from the Miami Isopycnic Coordinate Ocean Model (MICOM; Bleck and Smith, 1990; Bleck et al., 1992). The atmospheric component Community Atmosphere Model 4 (CAM4) has the option to use a modified chemistry-aerosol-cloud-radiation schemes developed for the Oslo version of CAM (e.g. CAM4-Oslo). The HAMburg Ocean Carbon Cycle (HAMOCC) model is adapted to the isopycnic model framework of NorESM and incorporated as the ocean biogeochemistry component of the model.





The basic evaluation and validation of the Climate Model Intercomparison Project Phase 5 (CMIP5) version of NorESM (NorESM1-M) is documented by Bentsen et al. (2013). Here we use a recently developed, computationally efficient variant of NorESM1-M: NorESM1-F (Guo et al., 2018), designed for multi-millennial and ensemble simulations while retaining the resolution ($2°$ atmosphere/land, $1°$ ocean/sea ice) and overall quality of NorESM1-M.

Compared to NorESM1-M, the model complexity of NorESM1-F is reduced by replacing CAM4-Oslo, that uses emissions of aerosols and explicitly simulates their life cycles, with the standard CAM4 that uses prescribed aerosol concentrations. The coupling frequency between atmosphere-sea ice and atmosphere-land is reduced from half-hourly to hourly, allowing the use of an hourly base time step for the sea ice and land components matching the radiative time step of the model; the dynamic sub-cycling of the sea ice is reduced from 120 to 80 sub-cycles. The last two changes result in a model speed-up of $\sim$30%

with a relatively small effect on the modeled climate. In addition, recent code developments in the ocean, atmosphere, and biogeochemistry components since NorESM1-M have been implemented as documented in detail by Guo et al. (2018). Of these code developments in NorESM1-F, the updated ocean physics in NorESM1-F lead to improvements over NorESM1-M in simulating the strength of the AMOC and the distribution of sea ice, both are important metrics in simulating past and future climates.

## 2.2   Experimental setup

In the MIS3 setup (Table 1), the solar constant is kept fixed at the pre-industrial value (1360.9 W m$^{-2}$), and the orbital parameters are set to values corresponding to 38 ka BP (Berger, 1978). In the Northern Hemisphere (NH), the chosen MIS3 time-slice shows enhanced insolation in spring (April-May-June) relative to present (Fig. 1), followed by reduced summer insolation (July-August-September). In the Southern Hemisphere (SH), changes in insolation are less pronounced, with stronger

fall (August-September-October) insolation and weaker winter (November-December-January) insolation.

    Concentrations of greenhouse gases (GHG) are set according to ice core measurements of typical interstadial conditions following Schilt et al. (2010): $CO_2$, $CH_4$, $N_2O$ concentrations are specified as 215 ppm, 550 ppb, and 260 ppb, respectively, which are identical to the MIS3 setup by Van Meerbeeck et al. (2009). Chlorofluorocarbon (CFCs) levels are set to zero.

    The ocean bathymetry is adapted based on an estimated sea level lowering of 70 m below present day (Waelbroeck et al.,

2002). As a consequence, many shallow ocean grid points on the shelf turn into land, thereby modifying the land-sea mask (Fig. 2). Most of the modifications occur in the northern high latitudes, e.g., the East Siberian Shelf, Laptev Shelf, and the Bering Strait (which is closed).

    The configuration of global ice sheet extent and height (Fig. 2) is derived from a data-constrained ice sheet model for 38 ka BP consisting of the Antarctic (Briggs et al., 2014), Greenland (Tarasov and Richard Peltier, 2002), North American (Tarasov

et al., 2012), and Eurasian (Lev Tarasov, personal communication) ice sheets. NorESM1-F does not have a dedicated land ice component, and an assumed ice sheet state during MIS3 compared to present day is taken into account by modifying the static land topography. The altitude of the land surface is kept at pre-industrial values outside the ice sheet areas. In the areas covered by ice, the maximum value of the original pre-industrial topography and MIS3 reconstructed ice sheet topography is used; this procedure prevents jumps to high topography adjacent to the ice sheet margin. The resulting MIS3 Laurentide Ice Sheet



reaches altitudes higher than 2500 m, but is significantly smaller, both in terms of ice extent and height, when compared to the LGM ice sheets such as ICE-6G (Peltier et al., 2015). The southeastern margin of the Laurentide Ice Sheet is further north and the surface is 300-900 m lower than during LGM. Similarly, the Eurasian Ice Sheet is smaller, but there is a significant amount of ice over Fennoscandia.

For the configuration of ice in the Barents Sea, care needs to be taken as this region is an important pathway allowing warm and saline Atlantic Water traveling north, therefore opening or closing it has significant consequences for Arctic ocean circulation and climate (Smedsrud et al., 2013). There is a lack of reliable geological evidence for the existence of ice in the Barents Sea before the LGM. However, sparse evidence from the Barents Sea-Svalbard region suggest there was little or no ice here during MIS3 (MANGERUD et al., 1998; Mangerud et al., 2008; Ólafur Ingólfsson and Landvik, 2013). Therefore,

the Barents Sea is kept open with a reduced water depth (by 70 m) in the MSI3 configuration of the model. The Canadian Archipelago is covered by ice, blocking the passage of water between Baffin Bay and the Arctic. In Antarctica, the Ross and Weddell Seas are covered by grounded ice rather than floating ice shelves as today.

The land surface vegetation type in the MIS3 configuration is set equal to the pre-industrial values, and the extra land points caused by sea level lowering are assigned as tundra (20% grass + 80% bare ground).

With the adjusted MIS3 land-sea mask and surface topography, a new river routing map is produced. For the ice free land surfaces, the river routing corresponds to the PI simulation. Where there is new land, due to the lower MIS3 sea level, the river mouths are extended to the ocean. For the ice covered areas, a new map is generated based on the ice topography, routing the water from the ice along the steepest gradient, either directly to the ocean, or to the nearest river if the ice margin terminates on land.

Salt equivalent to 0.6 g kg$^{-1}$ is uniformly added to the ocean, to account for the large amount of freshwater stored as ice on land. The vertical coordinate in MICOM is potential density with reference pressure of 2000 dbar, and is adapted from a present day range of 28.202-37.800 kg m$^{-3}$ to 28.672-38.270 kg m$^{-3}$ below the mixed layer, in order to account for the change in salinity and thus density. As for the PI experiment, the ocean model is initialised with modern temperature and salinity (Steele et al., 2001) with the above mentioned salinity increment applied. Note that no specific protocols on ocean initialisation are

defined for glacial simulations, e.g., within the Paleoclimate Modelling Intercomparison Project (PMIP), and groups choose to initialise their models either from present day conditions, or from a glacial ocean state. As illustrated by Zhang et al. (2013), equilibration time for the deep ocean can be significantly reduced (50% in their model) if a glacial rather than present day ocean state is utilised for initialisation. Note, however, that MIS3 is not a full glacial state, and has a climate in between LGM and PI. Model equilibration for the NorESM MIS3 experiment will be addressed and discussed in Section 3.1.

The MIS3 experiment was run for 2500 years and the PI experiment for 2000 years. When comparing the two, model years between 1800 and 2000 are averaged.



## 3 Results

### 3.1 Model spin-up

Both the PI and MIS3 experiments reach a quasi-equilibrium climate state after the multi-millennial integration, as indicated by the time series shown in Fig. 3. The differences between the global mean MIS3 and PI climate are summarised in Table 2.

In the following results, the statistical significance of the calculated trends are tested using the Student's $t$-test with the number of degrees of freedom, accounting for autocorrelation, calculated following Bretherton et al. (1999). Trends with $p$ values $< 0.05$ are considered to be statistically significant.

The MIS3 experiment exhibits a small negative TOA radiation balance (-0.16 W m$^{-2}$ averaged between model years 1801-2000 and -0.08 W m$^{-2}$ between 2301-2500; Fig. 3a). This results in a negative ocean heat flux at the surface, and cooling of the global ocean (Fig. 3d). The cooling trend is -0.05 °C per century over the model years 1801-2000, and decreases to -0.02 °C per century over the model years 2301-2500, both are statistically significant. Averaged between 1800 and 2000 model years, the global mean MIS3 ocean temperature is 1.7 °C colder than that of the PI experiment.

At the ocean surface, SSS in the MIS3 experiment exhibits negligible drift over the model years 1801-2000 (Fig. 3b,c), whereas SST shows a small statistically significant cooling trend of 0.04 °C per century. For MIS3, the global mean SST and SSS are 1.2 °C colder and 0.3 g kg$^{-1}$ saltier, respectively, compared to the PI experiment.

While the simulated MIS3 surface properties reach a quasi-equilibrium state, the ocean interior experiences a multi millennial cooling trend (Fig. 3d). The slow adjustment of the deep ocean is considered to be important for the evolution of ocean stratification and overturning circulation (Zhang et al., 2013; Marzocchi and Jansen, 2017). As a consequence, care should be taken when evaluating surface climatology, and deep ocean equilibration should be assessed. For the NorESM MIS3 simulation, we deem the aforementioned global mean ocean cooling trend to be small. For the deep ocean, the salinity trend in the Atlantic is found to be smaller than 0.006 g kg$^{-1}$ per century during the model years 1800-2000 (not shown), which is the threshold proposed by Zhang et al. (2013) for an ocean state of quasi-equilibrium.

Both experiments exhibit an increasing AMOC at the beginning of the model integration, followed by a gradual equilibration to a weaker state (Fig. 3e). The simulated pre-industrial AMOC at 26.5°N is 20.9 Sv, which is close to present day observations (~18 Sv; RAPID data from www.rapid.ac.uk/rapidmoc). For the first few hundred years of the MIS3 integration, the AMOC is significantly stronger than in the PI (by up to 10 Sv), but the difference between the two experiments is gradually reduced as the integration continues. Averaged between model years 1801-2000, the MIS3 simulated AMOC is 22.8 Sv, only 1.9 Sv greater than in the PI experiment.

Simulated time evolution of sea ice area, averaged over the northern and southern hemispheres, is shown in Fig. 4. Both experiments show a small drift at the end of the integration. In the NH, the MIS3 experiment shows a larger minimum sea ice area in September, and a smaller maximum sea ice area in March, compared to the PI experiment. The simulated smaller NH March sea ice area at MIS3, is caused by the large region in the peripheral areas of the Arctic Ocean being defined as land in MIS3, when sea level is lowered by 70 m (see Fig. 2). In the PI experiment, these areas are defined as ocean, which are ice covered during winter, resulting in a larger area of sea ice cover, even if the climate is warmer than at MIS3. As will





be shown later, sea ice in MIS3 has a larger extent in the NH when excluding the above offset due to changes in the land-sea mask. In the SH, MIS3 features larger sea ice area compared to PI in both seasons, even though the above land-sea effect is at play (mostly in the Weddell Sea and the Ross Sea). In austral winter (September), sea ice area continues to increase throughout the multi-millennial model integration. However, the trend is diminished towards the end of the simulation. In austral summer

(March), MIS3 sea ice area is comparable to PI and slowly increases until a sudden jump is detected close to model year 1600. After model year 1800 the simulated sea ice shows little drift in both experiments. The rapid increase in sea ice in the MIS3 experiment mainly arises from changes in the Weddell and Ross Sea areas of the model.

  The weakening of the AMOC, after the initial overshoot, occurs concurrently with a shoaling of North Atlantic Deep Water (NADW) and more intrusion of the Antarctic Bottom Water (AABW) as a manifestation of an adjustment of the deep ocean.

Previous studies related this behaviour to an expansion of Antarctic sea ice in a colder climate (Ferrari et al., 2014; Jansen, 2017): e.g., as Antarctic sea ice grows (Fig. 4), more brine is rejected, leading to more open ocean convection, favoring the formation of AABW. These processes will be further discussed in Section 5.1.

## 3.2 Simulated MIS3 climate

### Atmospheric surface temperature, circulation, and precipitation

Simulated annual mean surface air temperature change with respect to PI is shown in Fig. 5. Significant cooling occurs at high latitudes in both hemispheres. This is particularly clear above the MIS3 Laurentide Ice Sheet, where cooling reaches 25 °C, as well as in the Barents Sea and parts of the Nordic Seas, where sea ice expands (c.f. Fig. 13). The MIS3 experiment also exhibits noticeable cooling over the Kuroshio extension.

  Simulated global mean near surface air temperature during MIS3 is 2.9 °C cooler than the PI (Table 2). For comparison,

the cooling is smaller than the reconstructed LGM global mean cooling of 4.0 ±0.8 °C (Annan and Hargreaves, 2013) and the PMIP2 (Braconnot et al., 2007) and PMIP3 (five models; Braconnot and Kageyama, 2015) simulated LGM cooling range of 3.6-5.7 °C and 4.4-5 °C, respectively.

  The elevated surface of the MIS3 Laurentide Ice Sheet modifies the atmospheric stationary waves, rendering an enhanced, meandering wave pattern in the vicinity of the North American continent (Fig. 6); the displayed 500-mb geopotential height

in winter shows enhanced troughs in the northwestern Pacific and eastern Canada, and enhanced ridges in western Canada. A stronger, more zonal, and northward-shifted (by ∼4 degrees) subpolar jet above the North Atlantic is revealed by the 200-mb zonal wind (not shown). At the ocean surface, a deepened Aleutian low and the associated development of cyclonic surface wind is found in winter (not shown). The cyclonic wind anomaly advects warm air to Alaska and contributes to the reduced cooling as seen in Fig. 5. In the Atlantic, a southwestward migration of the Icelandic low and Azores high leads to broader,

stronger, and more southerly located westerlies over the North Atlantic (Fig. 7). The ice sheet induced wind anomalies over the North Atlantic are common features in the PMIP3 LGM simulations (Muglia and Schmittner, 2015). The zonal mean NH westerlies (surface zonal wind stress) increase by ∼20% relative to PI in NorESM, and shift equatorward by ∼4 degrees. Furthermore, the expansion of sea ice at MIS3 (see Fig. 13) induces a significant increase in the surface wind stress just off the





edge of the sea ice in the Nordic Seas and the Irminger Sea. In the Labrador Sea, a strong northerly wind anomaly is induced by the nearby Laurentide Ice Sheet.

In the tropics, the northeasterly trade winds are strengthened in the NH, while in the SH the southeasterly trade winds are relatively unchanged. In the Southern Ocean, the westerlies are strengthened in the Pacific Ocean sector and weakened in the Indian Ocean sector. The zonal mean of the westerly wind stress in the Southern Ocean shows a slight strengthening during MIS3 ($\sim$4%), with nearly no latitudinal shift.

Annual mean global precipitation decreases by 0.18 mm day$^{-1}$ as a consequence of the colder and more arid climate during MIS3. Geographically, during DJF, a significant decrease in precipitation is seen in the North Pacific, the western North Atlantic, the Barents Sea and in the Nordic Seas (Fig. 8). Greater precipitation is seen in the eastern and subtropical Pacific and in the eastern North Atlantic. Precipitation in the western Labrador Sea also increases, due to the reduced sea ice cover (see Fig. 13). In the tropics, precipitation in the African and South American monsoon regions, as well as in the western Pacific warm pool, is remarkably reduced, and a southward shift of the ITCZ occurs. In contrast, during JJA, western and northern Europe as well as the North Pacific features more precipitation; the Indian monsoon region experiences less precipitation, and in the tropics, the ITCZ moves south in the Pacific and north in the Indian Ocean.

## Ocean circulation and sea ice

Simulated global mean MIS3 SST is 1.2 °C colder with respect to the PI. The simulated cooling is comparable to the compiled MARGO reconstruction that estimates a SST cooling of 1.9±1.8 °C for the LGM (MARGO Project Members, 2009). The geographical distribution of the cooling (Fig. 9a) reflects the change in surface air temperature as shown in Fig. 5. The cooling is relatively modest (1-2 °C) in the tropical and subtropical oceans, and increases towards higher latitudes, in particular in the North Pacific, the Barents Sea, the Nordic Sea, and the Southern Ocean. In contrast, the central North Atlantic exhibits less cooling, and even exhibits warming near the center of the subpolar gyre and at the western rim of the Labrador Sea. While the "warm blob" in the subpolar gyre can be attributed to a shift of the North Atlantic Current at MIS3 (as suggested from the surface velocity and sea level fields; not shown), the relatively weak cooling in the NA subpolar region is likely caused by a stronger AMOC (Fig. 10) and a stronger and slightly more contracted subpolar gyre (Fig. 11) bringing more ocean heat from the tropics to this region (Fig. 12b) (e.g., the ocean heat transport increases by 15% at 40°N). The overall pattern of weak cooling in the North Atlantic during MIS3 can be compared to the recent twentieth century global warming, with a North Atlantic cooling anomaly, suggested to be caused by a reduction in the AMOC (Rahmstorf et al., 2015). The simulated warming along the western rim of the Labrador Sea (1-2 °C), apart from the contribution from a strengthened subpolar gyre (21% stronger compared to PI), is more directly related to the locally reduced MIS3 sea ice cover (Fig. 13a).

The cooling in the North Pacific is associated with a reduction of northward ocean heat transport in this region (Fig. 12b), e.g., ocean heat transport is 23 % smaller at 30°N during MIS3. In addition, the cooling is accompanied by expanded winter sea ice cover in the northwestern Pacific (Fig. 13a) and a southward shift of the North Pacific subpolar gyre (Fig. 11).

As northward ocean heat transport in the North Pacific decreases, southward ocean heat transport in the South Pacific and Indian Ocean increases (e.g., 13 % increase at 30°S). Further to the south, the MIS3 simulation shows enhanced meridional



ocean heat transport across the Antarctic Circumpolar Current (Fig. 12b). However, significant cooling is simulated in this region, and is associated with an equatorward expansion of sea ice (particularly in the western Indian Ocean sector; Fig. 13), concomitant with a northward shift of the Antarctic Circumpolar Current (not shown).

For the surface salinity (Fig. 9b), MIS3 is more saline as a result of the addition of 0.6 g kg$^{-1}$ salt into the global ocean due to expansion of ice on land. Exceptions are the North Pacific subpolar area, South Pacific subtropical area, South China Sea, eastern Atlantic, and off the Eurasian shelf. Strong salinity increases are found in Baffin Bay, the western Labrador Sea, the North American sector of the Arctic, and within and east of the Weddell Sea. The salinity decrease in the North Pacific and salinity increase in the Arctic and in Baffin Bay, can be partially attributed to the closure of the Bering Strait (Hu et al., 2010) at MIS3. The Bering Strait acts as a passage of freshwater from the Pacific to the Arctic Ocean, with an impact on salinity in Baffin Bay and the North Atlantic via the Canadian Arctic Archipelago. The Bering Strait, with a depth of ∼50 m at PI (and at present), is closed due to the lower sea level at MIS3, while the passage through the Canadian Arctic Archipelago is blocked by the presence of expanded MIS3 ice sheets (Fig. 2). A net volume transport of 1.3 Sv low salinity water, through the Bering Strait in the PI control run (present day observations are ∼0.8 Sv; Woodgate et al., 2005) is thus removed in the MIS3 run and contributes to the pattern described above.

Dramatic freshening takes place in the Eurasian sector of the Arctic at MIS3. Here, the Arctic river mouths extend further out to the open ocean, relative to the present day locations, due to the change of land-sea mask (Fig. 2). Such changes in the locations of river mouths, and thus freshwater input, lead to a decrease in salinity, including the region southwest of Norway, where a large glacial river is generated due to the presence of the nearby Fennoscandian Ice Sheet. In addition, the volume transport of saline water, of North Atlantic origin, into the Arctic via the Barents Sea Opening (from the southern tip of Svalbard to the northern tip of Norway), is reduced from 2.8 Sv at PI to 0.5 Sv at MIS3, contributing to the salinity decrease in the Eurasian sector of the Arctic.

The fresh surface water in the South China Sea is due to increased runoff during MIS3, whereas in the Southwest Pacific, surface freshening is due to a southward shift of the ITCZ and an overall decrease of evaporation minus precipitation in the region. Off the coast of Antarctica, enhanced formation of sea ice (Fig. 13), and the associated brine release, leads to an increase in surface salinity; the effect is especially pronounced in the Weddell Sea region.

**AMOC and Atlantic hydrography**

The AMOC redistributes heat and freshwater and plays a crucial role in the global climate. Our NorESM experiments show a strengthened AMOC at MIS3 (27.5 Sv) relative to the PI (24.3 Sv) (Fig. 10). The depth of the AMOC maximum for MIS3 is unchanged and located at 800 m. The vertical extent of the AMOC is deepened from 3500 m in the PI to 4200 m at 26° N (present day RAPID observations reveal a depth of ∼ 4300 m; Smeed et al., 2016). The deeper overturning stream function associated with AABW is contracted and weakened. For comparison, LGM simulations contributing to PMIP2 feature either deeper/stronger or shallower/weaker AMOC (Weber et al., 2007), whereas most PMIP3 LGM simulations exhibit a deeper/stronger AMOC relative to present day (Muglia and Schmittner, 2015).



Together with the changes to the AMOC, the deep Atlantic ocean exhibits changes in the distribution of water masses. The zonal mean Atlantic (including the Atlantic sector of the Southern Ocean and the Nordic Seas) temperature (Fig. 14a,b) shows that cooling occurs nearly over the entire water column in the Atlantic basin, with the strongest cooling detected near the bottom of the thermocline (>4 °C) and in the deep ocean below 3500 m (1.5-3 °C). The larger temperature decrease in the deep

North Atlantic compared to the South Atlantic suggests a more homogeneous deep water distribution in the Atlantic basin with a smaller inter-hemispheric gradient during MIS3 (Fig. 14b).

With more vigorous deep water formation in the NH (associated with a stronger AMOC) and also in the SH (as discussed later) during MIS3, a general enhanced upward motion of sea water is expected away from sinking regions (Munk, 1966); this leads to a thermocline that is displaced upwards with a sharper vertical gradient, contributing to the cold anomaly near the base

of the thermocline seen in Fig. 14b. Similar upward displacement of the thermocline and an associated cold anomaly are also seen in the Pacific Ocean (not shown). We further note that the especially cold anomaly centred around 30°N, 500-800 m depth (Fig. 14b) is primarily caused by the reduced warm Mediterranean outflow therein during MIS3 (not shown).

The Atlantic zonal mean salinity anomaly (Fig. 14c,d) shows an overall increase in salinity, except near the bottom of the pycnocline/thermocline, where the waters are subject to enhanced upwelling as discussed above. Greater freshening is also

observed in the saline Mediterranean outflow region where it is reduced during MIS3. For the deeper layers, there is a north-south asymmetry, with more saline bottom water in the South Atlantic ($\sim$0.4 g kg$^{-1}$) compared to the deep North Atlantic. Geological reconstructions of the glacial deep Atlantic hydrography from pore water measurements revealed a reversed north-south salinity at the LGM (Adkins et al., 2002), e.g., with AABW being more saline than NADW. In our MIS3 simulation, while the gradient is effectively lowered, the reduction is not sufficient to cause a salinity reversal.

The salinity effect dominates the change of density in the cold deep ocean, manifested by a larger increase of potential density in the Atlantic sector of the Southern Ocean (0.6-0.8 kg m$^{-3}$) than in the Atlantic (0.5-0.6 kg m$^{-3}$) (not shown). A comparison of ideal age distribution shows that ideal age in the South Atlantic and the Atlantic sector of the Southern Ocean is reduced by 200-400 years compared to that in the PI experiment (not shown). In the Southern Ocean, deep water masses are $\sim$400 years younger, and the entire water column shows a nearly homogeneous young water mass (<10 years), indicating

much enhanced ventilation of AABW.

**Sea ice**

As documented by Guo et al. (2018), the PI simulation of sea ice agrees well with observations in terms of both thickness and extent. The simulated MIS3 sea ice extent and the difference in thickness relative to the PI are shown in Fig. 13. In the NH, MIS3 sea ice is thicker by $\sim$2 m in the central Arctic in both March and September. In March, sea ice extends further

equatorward in the Pacific (reaching 40°N; not shown), and is associated with a cooling in the North Pacific (Figs. 5, 9a) and a southward migration of the Pacific subpolar gyre (Fig. 11).

In the Atlantic at MIS3, there is more winter sea ice south of Newfoundland and in the northeastern Labrador Sea (Fig. 13a). However, there is less sea ice in the western Labrador Sea, which is due to the strong northerly katabatic wind induced by the presence of the adjacent Laurentide Ice Sheet (Fig. 7). The Nordic Seas are partly ice-covered, with sea ice present off the





coast of Norway in winter. However, the central part of the Nordic Seas is ice free even in winter, due to the intrusion of warm Atlantic water across the Iceland-Scotland ridge. Note that the presence of winter sea ice, in particular in the region southwest of Norway, is not solely governed by the surface climate and inflow of Atlantic water; the simulated nearby river runoff from the Fennoscandian Ice Sheet contributes roughly 0.05 Sv of freshwater input (see also the SSS field in Fig. 9) to the region

(about a fourth of the Amazon River discharge), which favors the formation of winter sea ice (Fig. 13a).

In September, the simulated MIS3 sea ice retreats and nearly coincides with PI sea ice extent in the Pacific side (not shown) and in the Labrador Sea (Fig. 13b). Greater sea ice extent is found along the coast of South Greenland and in the Nordic Seas. The Barents Sea is fully ice-covered also in summer at MIS3, in contrast to the PI experiment where this region is seasonally ice-free.

In the SH, MIS3 shows extended Antarctic sea ice cover in both seasons. The seasonal cycle is large in both MIS3 and PI experiments, with the total sea ice area varying by a factor of 3 and 4 between March and September, respectively (Fig. 4; Table 2). Furthermore, sea ice is thicker during MIS3, especially in the Weddell Sea region.

**Modes of variability**

In this section, we briefly evaluate the change of two important climate internal variabilities: the El Niño-Southern Oscillation

(ENSO) and the Northern Annular Mode (NAM).

The tropical Pacific cools nearly uniformly during MIS3, with a small change in the zonal SST gradient in the eastern Pacific cold tongue and western Pacific warm pool region (Fig. 9a). The amplitude of the SST change in the NINO3.4 region (170°W-120°W, 5°S-5°N) during MIS3 is about 1.2 °C relative to PI, with a weak seasonal cycle (Fig. 15a). As a measure of ENSO, the standard deviation ($\sigma$) of the detrended monthly SST anomalies in the NINO3.4 region is smaller during MIS3 (0.45 °C)

relative to PI (0.58 °C). The reduction is across all months and is greater in boreal autumn and winter (Fig. 15b), leading to a slightly weakened annual cycle of NINO3.4 SST variability. Similar reductions of ENSO variability were reported in CCSM3 and CCSM4 LGM simulations (Otto-Bliesner et al., 2006; Brady et al., 2013), but the results show disagreements among the PMIP2 LGM model simulations (Zheng et al., 2008).

MIS3 shows small and negative skewness across most of the year (Fig. 15c); the annual cycle is smaller during MIS3, with

the largest discrepancy in boreal summer compared to PI. For the frequency of ENSO events, the NINO3.4 index exhibits most power over 2-5 years for both MIS3 and PI experiments, with the former showing more power in the lower and the latter in the higher end of the range.

Fig. 16 shows the composite anomalies of DJF SST during El Niño years for the MIS3 and PI experiments. An El Niño year is defined here as a year with the NINO3.4 SST anomalies greater than $1.5\sigma$ for three consecutive months, with at least

one DJF months. The SST anomalies in the tropical Pacific are weaker during MIS3 and have a smaller westward extent. The maximum SST anomalies, centered around 120°W, are ~1.2 °C at MIS3, compared to ~1.5°C in PI. Stronger negative SST anomalies at MIS3 are seen in the subtropical Pacific in both hemispheres. In the southern Indian Ocean, stronger positive SST anomalies are simulated during MIS3, whereas the anomalies are weaker in the eastern and central Indian Ocean relative to PI.



The NAM (also known as the Arctic Oscillation) is defined here as the first empirical orthogonal function (EOF) of the NH (20-90° N) DJF sea level pressure (SLP) anomalies. The NAM-explained winter SLP variance is reduced during MIS3 (27%) relative to PI (30%), with the centre of action over the Arctic slightly weaker and slightly eastward-shifted (Fig. 17a,b). The shape of the EOF pattern is more asymmetric over the Arctic at MIS3. With the presence of large ice sheets during MIS3, the
simulated centre of action is weakened relative to PI in the North Pacific and eastern North Atlantic, resulting in a gradient from the mid-latitude to the pole that is weaker during MIS3.

## 4   MIS3 simulation forced by stadial conditions

The NorESM MIS3 simulation presented above is representative of an interstadial climate, i.e. a relatively warm period during the last glacial; in agreement with paleo reconstructions, this includes, Greenland temperatures only 5–8 °C colder than PI
(Huber et al., 2006), a strong AMOC (Henry et al., 2016), and enhanced sea ice-cover in the Nordic Seas (Rasmussen and Thomsen, 2004; Dokken et al., 2013). In particular, a high resolution reconstruction of MIS3 sea ice, based on analysis of biomarkers from a marine sediment core in the south Nordic Seas (Sadatzki et al., manuscript submitted to Science Advances), finds near-perennial sea ice cover during cold stadials, and ice free periods during warm interstadials. This supports our modeled sea ice distribution and suggests a simulated MIS3 insterstadial climate state. However, questions remain as to the state of the
cold stadial climates of MIS3, which is characterized by much colder Greenland temperatures, a completely ice-covered Nordic Seas, and reduced AMOC compared to the present day (based on the same references as above). These changes are reinforced during stadials including Henrich-events, when the AMOC shows a further weakening owing to the massive release of icebergs into the North Atlantic (e.g., Henry et al., 2016).

It is unclear if the baseline climate during MIS3 is a stadial or interstadial state, nor is it clear that there is indeed a baseline
climate, as the climate states can be inherently oscillatory (Peltier and Vettoretti, 2014; Brown and Galbraith, 2016; Klockmann et al., 2018). To examine the possibility of simulating a cold MIS3 stadial climate with NorESM, an additional experiment is branched off from the MIS3 interstadial control experiment after 1700 model years. The MIS3 "stadial" experiment is run for 800 years with 40 ka orbital forcing and reduced GHG levels (e.g. $CO_2$ - 200 ppm, $CH_4$ - 450 ppb, $N_2O$ - 220 ppb) according to Schilt et al. (2010). The major difference compared to the interstadial experiment is that in the "stadial" experiment the CO2
is lowered by 15 ppmv.

In the MIS3 "stadial" experiment, the global near surface temperature cools by 0.4°C relative to the MIS3 interstadial experiment (Fig. 18). The greatest cooling occurs along the newly formed sea ice margin in the Nordic Seas (∼3.8°C) and southwest of the Bering Sea (∼1.8°C). The change in South Greenland is ∼1.3°C. The zonal mean cooling is ∼1°C in the NH high latitudes and is reduced to ∼0.3°C in the tropical and subtropical regions (figure not shown); in the Southern Ocean, the
zonal mean cooling is ∼0.7°C, decreasing to ∼0.4°C over Antarctica.

The surface cooling in the Nordic Seas in the "stadial" experiment is reflected in the growth of sea ice in the region, e.g. winter sea ice slightly expands southwards in the Nordic Seas (Fig. 18). Similar increase of sea ice is also simulated in the northwestern Pacific, whereas in the Labrador Sea and in Antarctica, sea ice distribution is nearly identical in the two





experiments. Furthermore, there is a negligible change to the AMOC in the "stadial" experiment. Together, these results indicate that given the changes to the GHG levels that is typical of a stadial state, a cold Greenland climate with a weak AMOC cannot be reproduced in our MIS3 setup.

# 5 Discussion

## 5.1 Simulated AMOC in MIS3

Abrupt climate changes such as D-O events have been shown to involve changes in both the geometry and strength of the AMOC, as indicated by a number of marine proxy reconstructions (e.g., see the review by Lynch-Stieglitz, 2017) and numerical simulations (e.g., Peltier and Vettoretti, 2014; Brown and Galbraith, 2016). A stronger and deeper AMOC during MIS3 interstadial is simulated compared to the PI (Fig. 10). Given its crucial role in the MIS3 climate, we further discuss the simulated AMOC and the associated distribution of NADW and AABW in this section.

Previous studies have argued that the increased AABW ventilation and production during glacial times is driven by expanded Antarctic sea ice and enhanced brine rejection during ice formation (Shin et al., 2003; Ferrari et al., 2014): the brine induces a negative buoyancy flux, increasing open ocean convection and leading to enhanced formation of highly saline AABW (Fig. 14d). Jansen (2017) shows that the changes in the deep ocean stratification and circulation can be interpreted as a direct consequence of atmospheric cooling during glacial times, which induces Antarctic sea ice growth and initiates the processes described above.

The simulated enhanced ventilation of AABW during MIS3 is not comparable with studies of the LGM, during which benthic foraminiferal $\delta^{13}$C data suggests that AABW dominated the water column in the Atlantic below $\sim$2 km (Curry and Oppo, 2005), together with a shallower NADW cell. However, measurements of $^{231}$Pa/$^{230}$Th, in combination with $^{143}$Nd/$^{144}$Nd ($\epsilon_{Nd}$), indicate that a strong AMOC existed at LGM despite of a shallow upper cell (Böhm et al., 2015). Böhm et al. (2015) also show that an active AMOC, neither weaker nor shallower than present day, prevailed over the last glacial cycle, including the D-O interstadials (the exceptions are Heinrich stadials exhibiting a weaker and shallower NADW cell). Our simulated MIS3 interstadial AMOC agrees with these reconstructions based on chemical tracers, and the difference with the AMOC at LGM can be partly (if not fully) attributed to less Antarctic sea ice formation due to the MIS3 climate being milder than that at the LGM.

While deep water production in the Southern Ocean has the potential to displace and reduce the strength of the NADW cell, competing effects are at play in the North Atlantic. For example, the altered surface westerlies in the North Atlantic caused by the elevated Laurentide Ice Sheet (Figs. 2,7) are shown to be able to induce a deeper and stronger AMOC by transporting more salt northward within an intensified gyre circulation, favouring deep ocean convection (Muglia and Schmittner, 2015; Klockmann et al., 2016); the closure of Bering Strait leads to an increase of surface salinity in the North Atlantic thereby invigorating deep ocean convection and strengthening the AMOC (Hu et al., 2010). To isolate and assess the relative impact of these different processes requires a suite of dedicated sensitivity studies which is beyond the scope of this paper, but it is worth mentioning that the processes that take place in both hemispheres act together to create the AMOC shown in Fig. 10.





During the last glacial, sea level lowering and the removal of shallow continental shelves (Fig. 2) result in enhanced tidal dissipation in the open ocean (Egbert et al., 2004), implying enhanced deep ocean mixing and a strengthened AMOC. Schmittner et al. (2015) demonstrated that such tidal effects can dominate over surface buoyancy effects and lead to a strengthening of AMOC by ∼40%. Such tidal effects are not considered in the current study, but once included, the AMOC is expected to
strengthen and deepen, potentially displacing the lower AABW cell in the Atlantic.

## 5.2   MIS3 sensitivity to $CO_2$ and ice sheet size

With the multi-millennial long integration of the MIS3 simulation presented in this work, only one stable climate state is found, and the model reaches a quasi-equilibrium with a small drift towards the end of the integration. In this section, we explore the potential for model bi-stability associated with the transition between the warm interstadial and cold stadial climate states of
MIS3. We do so by perturbing the model with changes in atmospheric $CO_2$ concentrations and the size of the Laurentide Ice Sheet.

Our NorESM MIS3 simulations agree with that of Van Meerbeeck et al. (2009) using the LOVECLIM model: given MIS3 boundary conditions, the interstadial climate is the equilibrium climate state, whereas the stadial climate is a perturbed state. A $CO_2$ change of 15 ppmv (typical of stadial conditions) is not sufficient to induce transitions between the two states. However,
Zhang et al. (2017) reported that a gradual change of $CO_2$ concentrations by 15 ppmv in their experiment with intermediate glacial conditions can be sufficient to trigger transitions between weak/strong AMOC modes and therefore stadial/interstadial states. In contrast, Klockmann et al. (2016) examined the effect of $CO_2$ changes on the AMOC strength and geometry in an LGM setup, and found a weakening and shoaling of AMOC with decreasing $CO_2$ (from 284 to 149 ppmv), but without transitioning into a weak AMOC mode. They argued that the presence of LGM ice sheets could enhance the AMOC by
impacting the surface wind stress in their model, and thus help maintain the stability of the AMOC. Interestingly, Klockmann et al. (2018) later reported that with a PI ice sheet configuration, the AMOC exhibits bi-stability with D-O like oscillatory behaviour at a $CO_2$ level of 206 ppmv, above (below) which a strong (weak) AMOC mode persists.

The studies by Klockmann et al. (2016, 2018); Zhang et al. (2017) on the sensitivity of climate states to $CO_2$ levels and ice sheet sizes, together with the studies of (e.g., Zhang et al., 2014; Brown and Galbraith, 2016; Galbraith and de Lavergne, 2018),
point to the possibility that there could exist a certain range of $CO_2$ levels and ice sheet sizes in which the model is subject to a mode transition and even excitation of self-oscillatory behaviour in North Atlantic climate. The existence and range of such a "window" of $CO_2$ levels and ice sheet heights, however, is expected to be model-dependent. In order to explore the potential for model bi-stability with NorESM and the existence of a cold stadial state given MIS3 boundary conditions, we investigate the model response to large variations of $CO_2$ and ice sheet height, using the interstadial simulation as the baseline experiment.
Another motivation for seeking a cold climate is that, as discussed by Peltier and Vettoretti (2014), a cold Heinrich stadial-like state is required to "kick" the system into a self-oscillatory behaviour. Even without the self-oscillation, a cold stadial state would serve as a useful baseline experiment for investigating potential triggers of the abrupt cold-to-warm D-O transitions.

It is not within the scope of this paper to perform a thorough examination of the model response to every combination of $CO_2$ and ice sheet changes. Rather, we report the model response in the more extreme cases, with a primary focus on the



response of the AMOC and sea ice. For the $CO_2$ sensitivity experiments, we reduce the values from 215 ppmv to 180 ppmv and 140 ppmv, respectively. For the ice sheet sensitivity experiments, we reduce the height of the Laurentide Ice Sheet by 50% and 100% (the latter is equal to the pre-industrial orography), while keeping the ice mask unchanged. All the sensitivity experiments are branched off from the MIS3 interstadial simulation, and all other parameters are kept fixed.

Contratry to the studies cited above, the NorESM MIS3 experiments exhibit surprising stability without any significant changes in Greenland temperature, sea ice, or AMOC (Fig. 19). For the low $CO_2$ experiments, winter sea ice expands slightly in the Nordic Seas, without any notable changes in the strength and regions of convection (not shown). As a consequence, the AMOC remains strong (Fig. 19). For the experiments with a reduced Laurentide ice sheet, surface wind stress fields are altered (mainly shifted northwards in the North Atlantic; not shown), whereas the strength of AMOC is not affected.

The results of the sensitivity experiments to $CO_2$ and ice sheet height underscore the question of model dependence on the background climate in simulating AMOC transition/bi-stability. In our simulation where the Labrador Sea and the Norwegian Sea are the major convection sites, a significant change of ocean circulations would not be expected unless the Labrador Sea and the Norwegian Sea are covered by ice inhibiting convection. However, the NorESM MIS3 and PI experiments both appear to be in a stable regime with a strong AMOC and strong convection in the Labrador and Norwegian Seas. As a consequence,

the model state is relatively distant from a potential threshold, including a bi-stability of the AMOC and sea ice. In addition, the NorESM1-F model features a low climate sensitivity (the equilibrium climate sensitivity in response to a doubling of atmospheric $CO_2$ is 2.3 °C) which plays a role in the relatively weak response of the MIS3 climate to a further $CO_2$ decrease.

To trigger a cold stadial-like climate state in the NorESM, other mechanism including enhanced ice berg calving and freshwater input to the North Atlantic from the Laurentide, Greenland and Fennoscandian ice sheets should be considered. There

is a rich literature on applying freshwater flux of different magnitudes and locations to study climate response and transitions (e.g. Stouffer et al., 2006; Roche et al., 2010). While beyond the scope of this study, the prospects of perturbing the MIS3 interstadial climate with freshwater fluxes will be explored in detail in a companion study.

## 6   Conclusions

In this paper, we present an equilibrium simulation of Marine Isotope Stage 3 forced by 38 ka BP boundary conditions, with a

recently developed version of the NorESM featuring a horizontal resolution of 2° atmosphere/land and 1° ocean/sea ice. The fast performance of the model allows the experiments to be integrated for 2500 years. The boundary conditions are notably different from the pre-industrial mainly in orbital forcing, greenhouse gases, and the height and extent of the global ice sheets.

The reported simulation, with its current length of integration, does not produce spontaneous transitions between colder stadial and warmer interstadial climate states. Rather, we obtain a MIS3 background climate state with a state-of-the-art climate

model that can serve as a baseline for investigating mechanisms behind D-O event by discriminating different factors that can invoke abrupt transitions, e.g., freshwater input, changes in GHG concentrations, ice sheet size, orbital forcing, and ocean diapycnal mixing.



Despite a small drift due to the ocean cooling as the model is integrated, the model reaches a quasi-equilibrium state in terms of both surface metrics and deep ocean hydrography. We analyze the large-scale features of the mean climate states and the model internal variabilities, and compare the results to previous simulation studies of both MIS3 and LGM climates. The major findings are as follows:

– Simulated MIS3 interstadial climate is globally 2.9 °C cooler relative to the PI, with amplified cooling in the high latitudes, especially above the ice sheets and near the edges of the newly formed sea ice. The presence of the Laurentide Ice Sheet amplifies the atmospheric stationary waves, leading to an enhanced and northward-shifted jet stream, and stronger and southward-shifted wind stress at the ocean surface. Global mean precipitation during MIS3 is 0.18 mm day$^{-1}$ lower in the colder MIS3 climate, with both seasonal and geographical changes including a southward shift of the 10 ITCZ.

– The global mean SST at MIS3 is 1.2 °C colder than that at PI, with a pattern of modest cooling in the tropics and enhanced cooling at high latitudes. The North Atlantic subpolar region is characterised by less cooling owing to enhanced AMOC and ocean heat transport. Greater cooling is simulated in the North Pacific associated with the expansion of sea ice and southward shift of the subpolar gyre.

– Despite the uniform addition of salt (by 0.6 g kg$^{-1}$) into the global ocean, the distribution of SSS exhibits an inhomogeneous pattern of both salinity increase (e.g., in the central Arctic, the Baffin Bay, and the Weddell Sea region) and decrease (e.g., off the Eurasian shelf, in the North Pacific, and western Pacific and the South China Sea). The closure of the Bering Strait, the increase in sea ice formation, and the change of glacial river routing all play important roles in determining the SSS pattern.

– The upper cell of the AMOC is deepened and intensified under the influence of competing factors from both hemispheres: e.g., the cutoff of freshwater input due to the closed Bering Strait and the strengthened surface wind stress in the NH subpolar region, both tend to invigorate AMOC. In the Southern Ocean, expansion of Antarctic sea ice stimulates AABW production by enhanced salt rejection and deep water production during sea ice formation. The results are supported by marine proxy records indicating an AMOC comparable to the present day during the last glacial (except during Heinrich 25 stadials). The enhanced deep ocean ventilation in the Atlantic sector of the Southern Ocean leads to reduced (but not reversed) deep ocean north-south salinity gradients in the Atlantic. The Atlantic displays pronounced cooling below 3000 m in both hemispheres and near the base of the thermocline, the latter due to stronger upwelling of deep water as a result of enhanced deep water formation in both hemispheres. Reduced Mediterranean outflow during MIS3 contributes to the notable cooling and freshening observed around 30N, 500-800 m depth.

– Sea ice is notably thicker and greater in extent during MIS3 in both hemispheres and seasons. Arctic sea ice is about 2 meters thicker and extends further equatorward in the Pacific during winter. The Nordic Seas are partly ice-covered in boreal summer; in winter, sea ice extent is greater, but includes an opening in the south due to the intrusion of warm

Atlantic Water. In the Southern Hemisphere, Antarctic sea ice is thicker (mainly in the western Indian Ocean sector) and extends further north.

– Simulated ENSO variability is weakened in the MIS3 compared to the PI simulation. For the Arctic Oscillation, simulated centres of action over the Arctic and North Pacific are both weakened, with the latter much reduced due to the presence of the elevated Laurentide Ice Sheet.

– A sensitivity experiment with boundary conditions typical of MIS3 stadial conditions does not reproduce the cold temperatures observed on Greenland, indicating that the interstadial climate in NorESM is relatively stable, and forms the baseline climate during MIS3. Further sensitivity experiments including large changes in atmospheric $CO_2$ levels and Laurentide Ice Sheet heights, aimed at perturbing the system into a cold stadial-like climate, show that the model is not subject to any bi-stability of the AMOC or sea ice. This underscores the role of model dependence in studying abrupt climate transitions during MIS3 or in other geological periods in general.

*Code and data availability.* The model code can be obtained upon request. Instructions on how to obtain a copy are given at $https://wiki.met.no/noresm/gitbestpractice$. The full set of model data will be made publicly available through the Norwegian Research Data Archive at $https://archive.norstore.no$ upon publication.

*Competing interests.* The authors declare that they have no conflict of interest.

*Acknowledgements.* This work was funded by the Ice2Ice project that has received funding from the European Research Council under the European Community's Seventh Framework Programme (FP7/2007-2013)/ERC grant agreement No. 610055. We acknowledge Peter Langen, Christian Rodehacke, and Will Roberts for the discussion of configuring MIS3 boundary conditions during a workshop in May 2016. We thank Lev Tarasov for providing the ice sheet data to force the model. We also thank Anne-Katrine Faber for commenting on part of the manuscript. The simulations were performed on resources provided by UNINETT Sigma2 - the National Infrastructure for High Performance Computing and Data Storage in Norway (nn4659k, ns4659k).




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





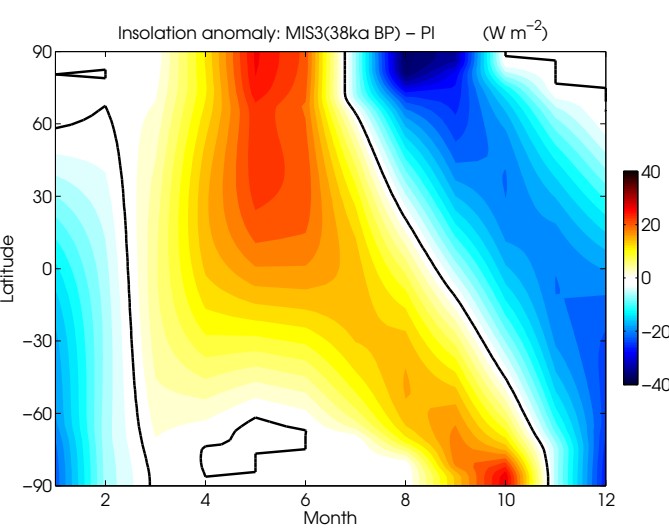

**Figure 1.** The insolation anomalies of MIS3 at 38 ka BP relative to present day (W m$^{-2}$). The thick contour is the zero isoline.



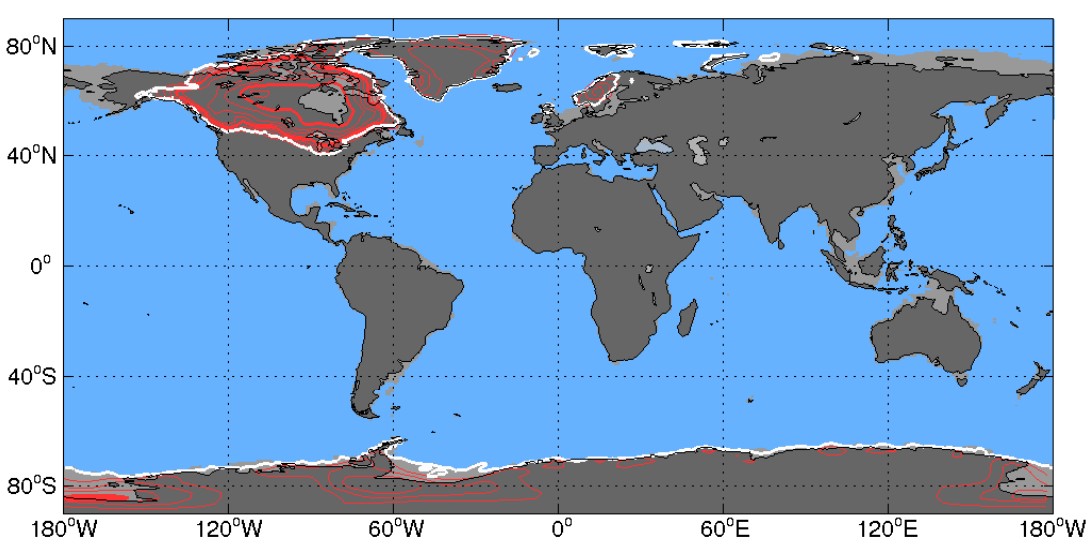

**Figure 2.** Land-sea mask (light grey/blue shading; dark grey are modern land), ice sheet extent (while line) for the MIS3 experiment, and the difference of ice sheet orography relative to PI (red contours with an interval of 250 m; the 1000 and 2000-m isolines are highlighted with bold red lines).





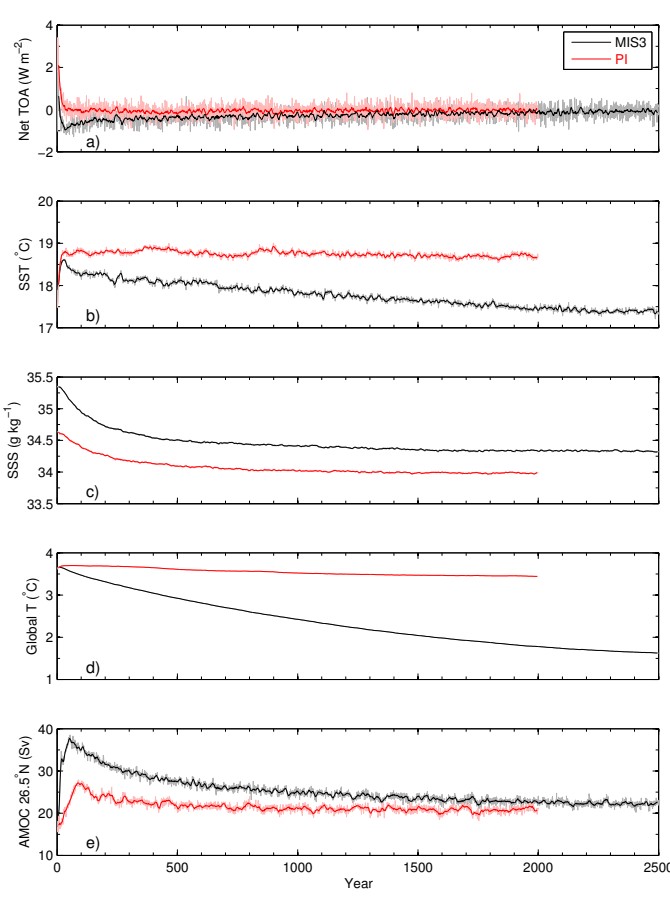

**Figure 3.** Time series of a) TOA radiation balance, b) SST, c) SSS, d) global mean ocean temperature, and e) AMOC strength at 26.5° N for the MIS3 (black) and PI (red) experiments. Light colors denote annual mean values, and dark colors denote 10-year running mean values.





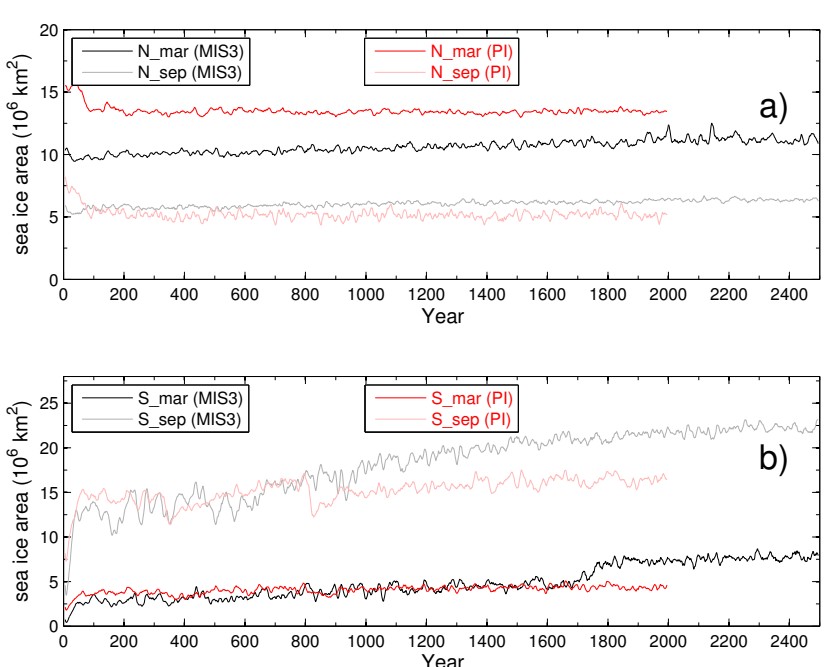

**Figure 4.** Time series of a) Northern Hemisphere and b) Southern Hemisphere sea ice area for the MIS3 (black) and PI (red) experiments. The data shown are 10-year running mean values.



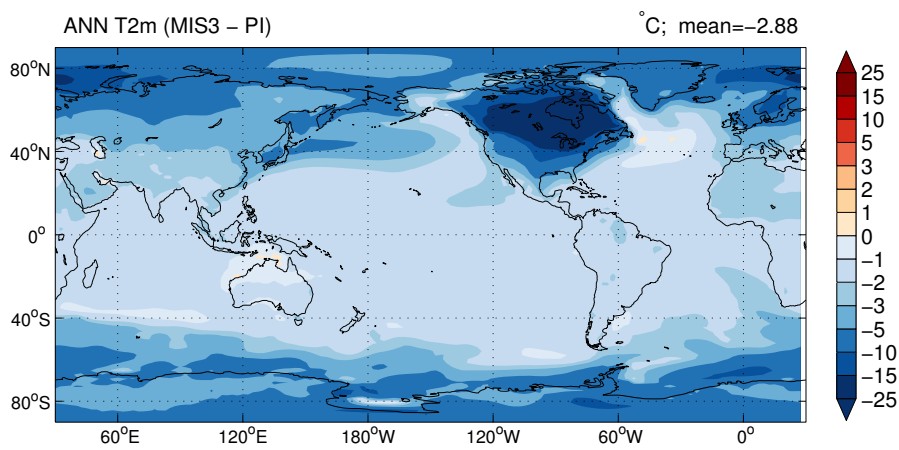

**Figure 5.** Simulated MIS3 minus PI annual mean near surface temperature (°C).

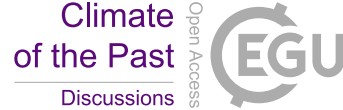



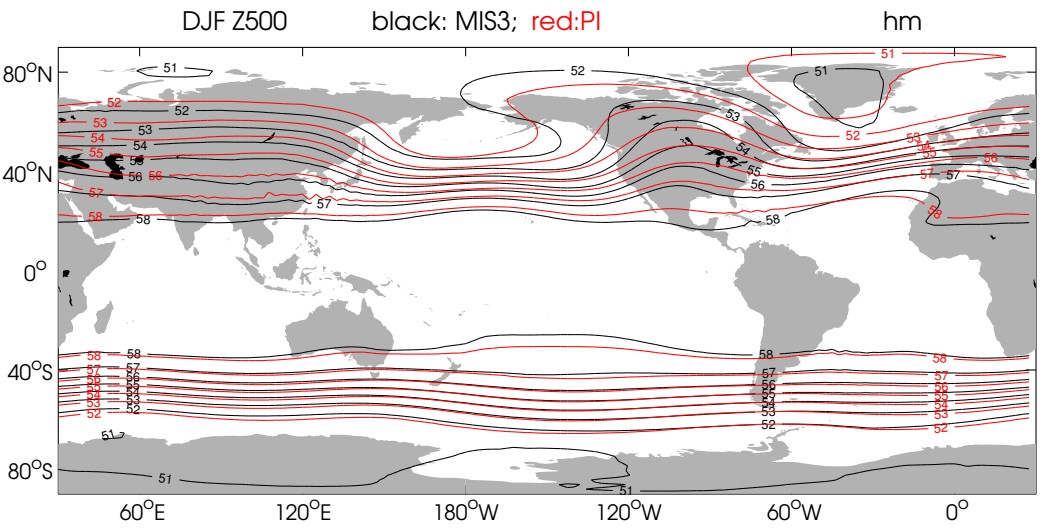

**Figure 6.** Simulated MIS3 and PI DJF 500-mb geopotential height (hm).





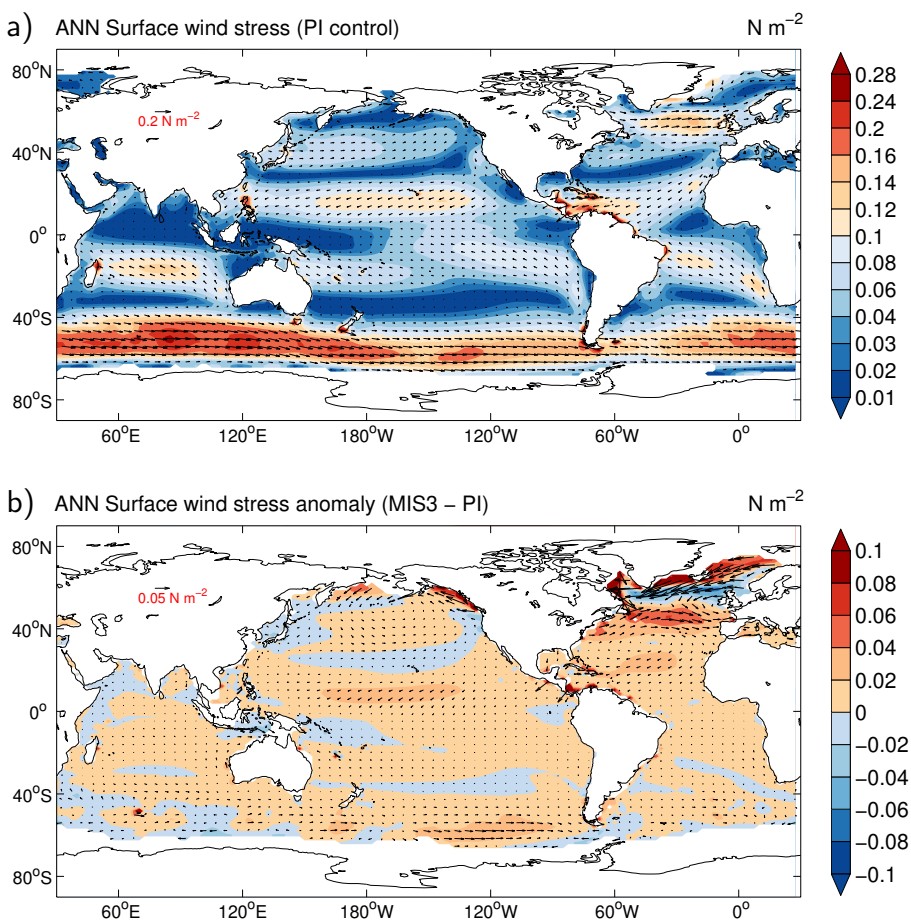

**Figure 7.** Simulated annual surface wind stress over the ocean for a) PI and b) MIS3 minus PI (N m$^{-2}$).





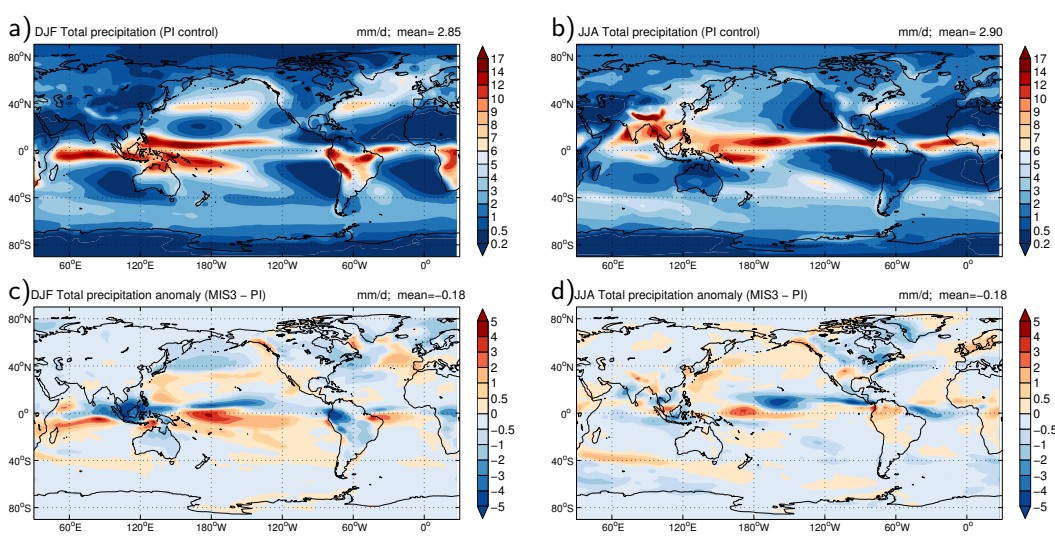

**Figure 8.** Simulated seasonal (DJF/JJA) total precipitation for a,b) PI and c,d) MIS3 minus PI (mm day$^{-1}$).





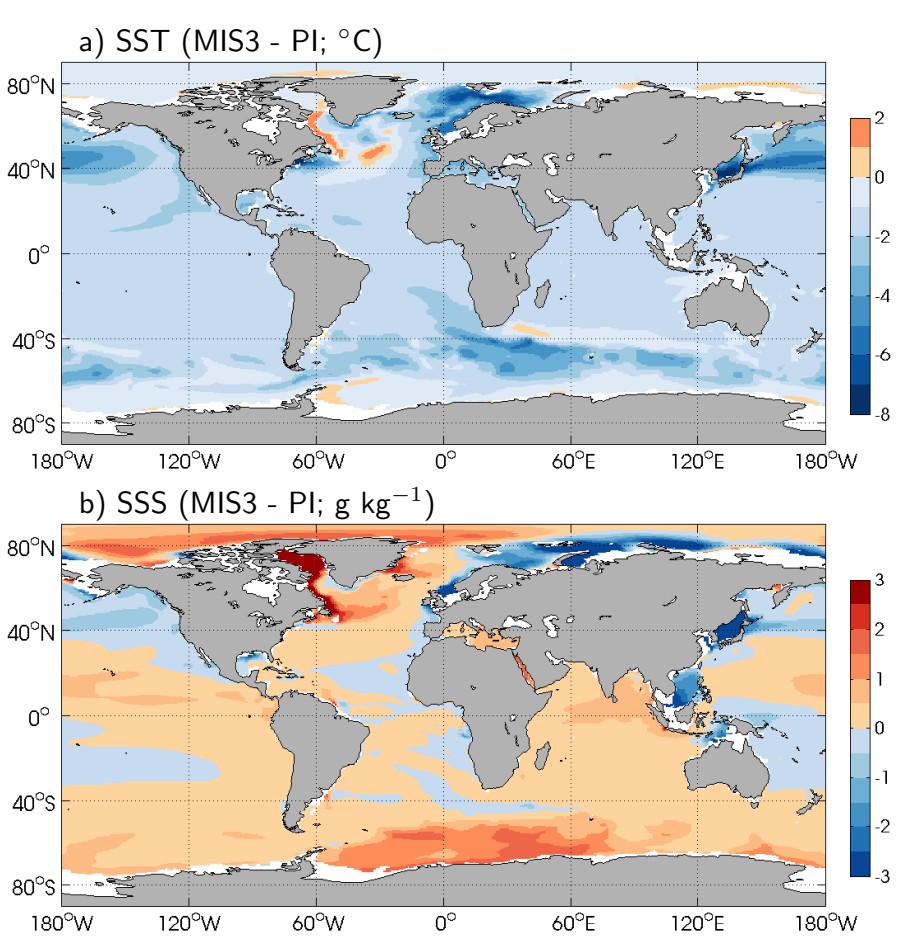

**Figure 9.** Simulated difference of MIS3 a) SST and b) SSS relative to PI.



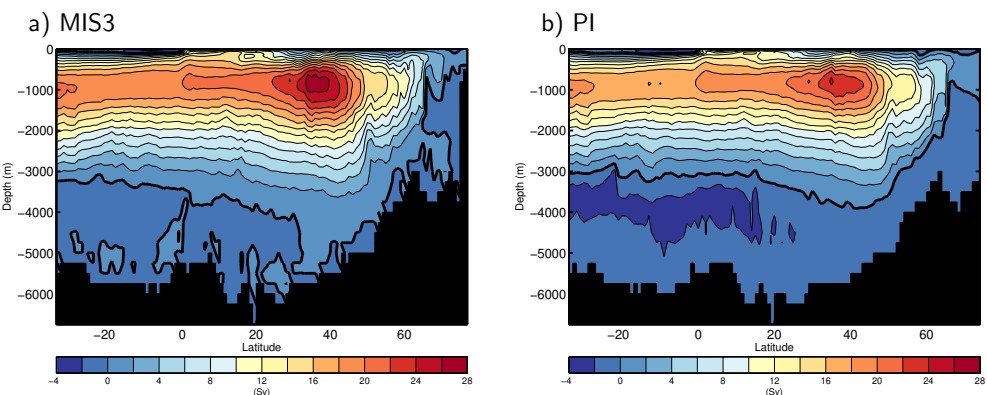

**Figure 10.** Stream functions of AMOC for a) MIS3 and b) PI simulations. The thick black line denotes the zero contour line.



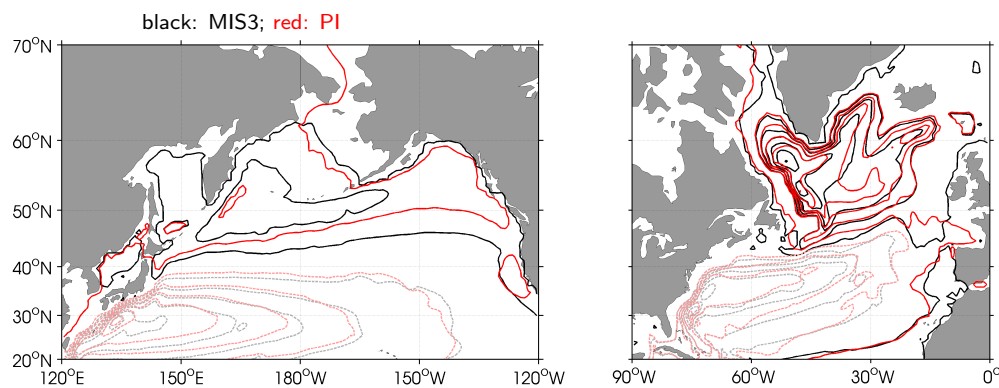

**Figure 11.** Barotropic stream functions of ocean circulation in the North Pacific and North Atlantic region for MIS3 (black) and PI (red) simulations. Contour intervals are 10 Sv, with solid and dashed colors denoting positive and negative values, respectively.





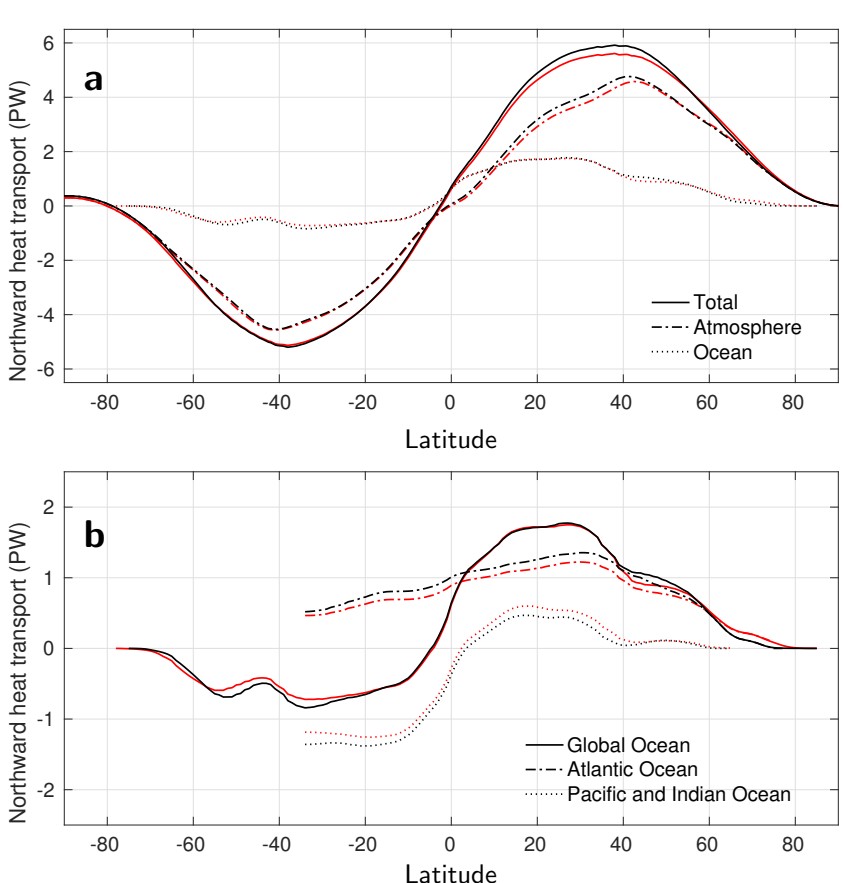

**Figure 12.** Simulated MIS3 (black) and PI (red) northward heat transport for a) global atmosphere, ocean, and total, and for b) global ocean, Atlantic Ocean, and Pacific and Indian Ocean. The ocean heat transport is calculated directly from the ocean model, and the atmospheric heat transport is calculated by meridional integration of the difference between the zonal integration of the net TOA and surface heat flux.





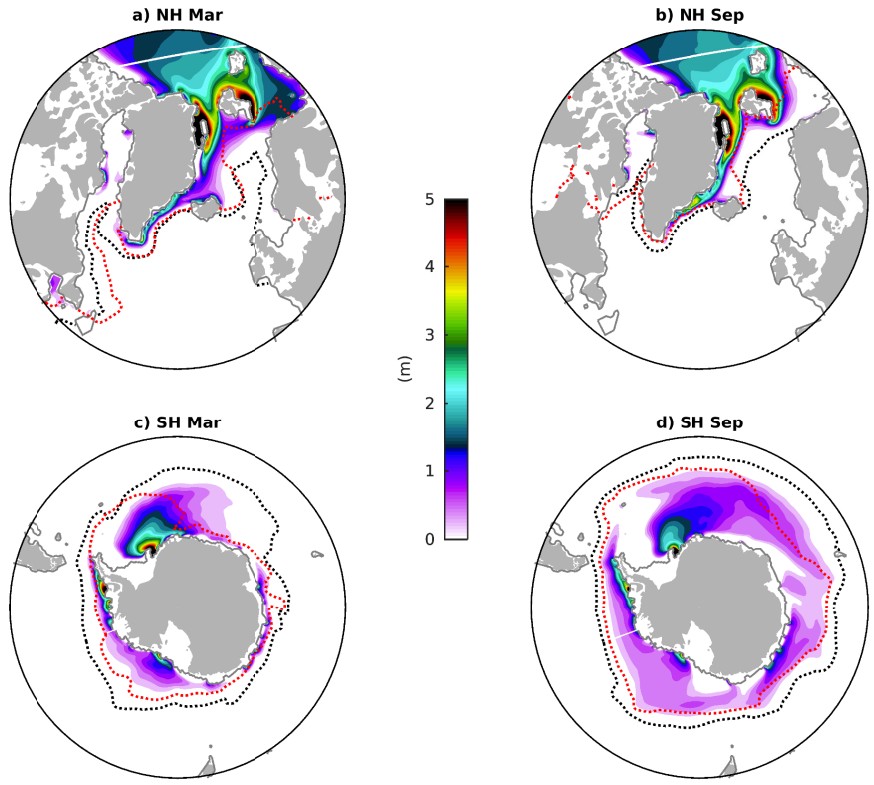

**Figure 13.** Simulated difference of MIS3 sea ice thickness with PI (shading; m) for a) NH March, b) NH September, c) SH March, and d) SH September. Black and red dashed lines denote the 15% sea ice concentration for MIS3 and PI, respectively. The grey line is the MIS3 coast line.

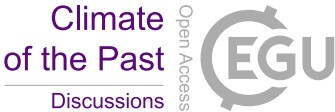

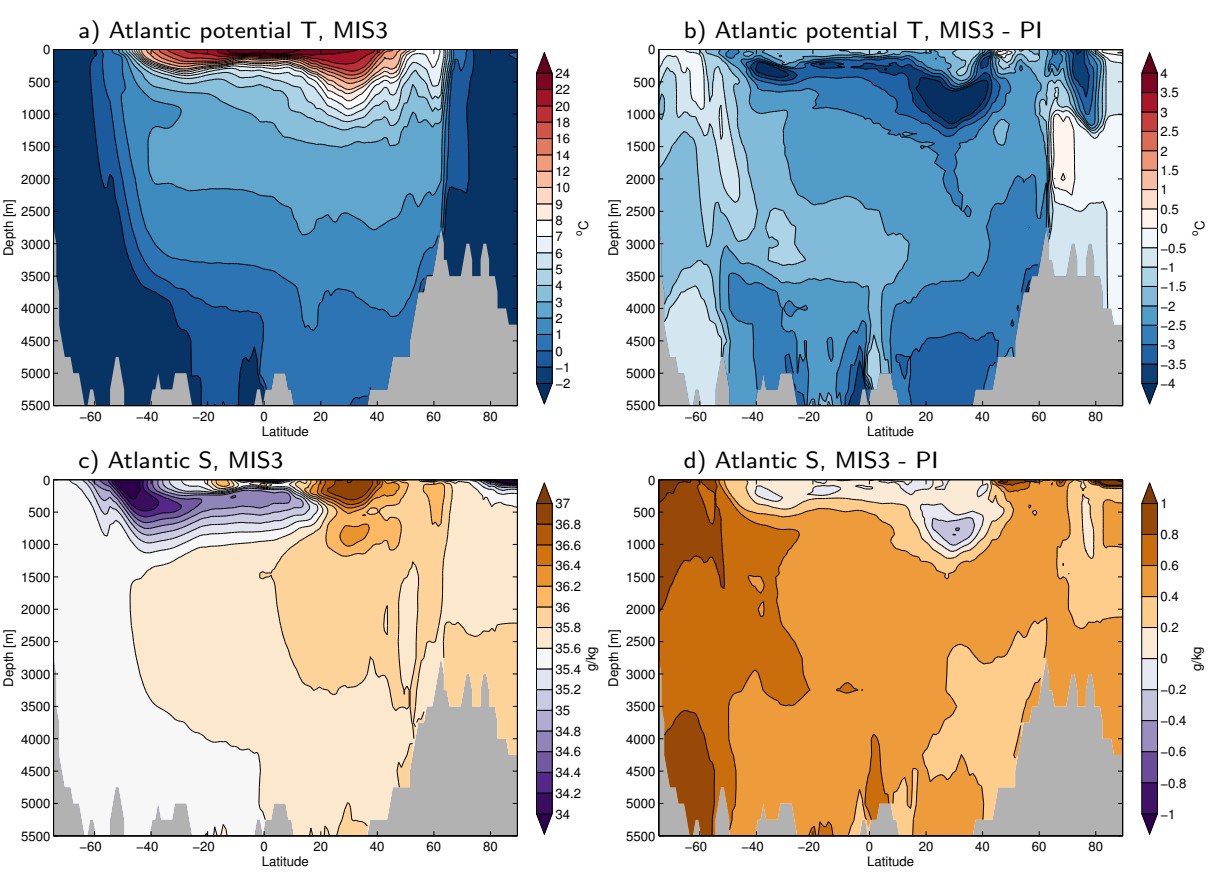

**Figure 14.** Atlantic zonal mean distribution of a) potential temperature, c) salinity for the MIS3 experiment, and b,d) the differences relative to PI.





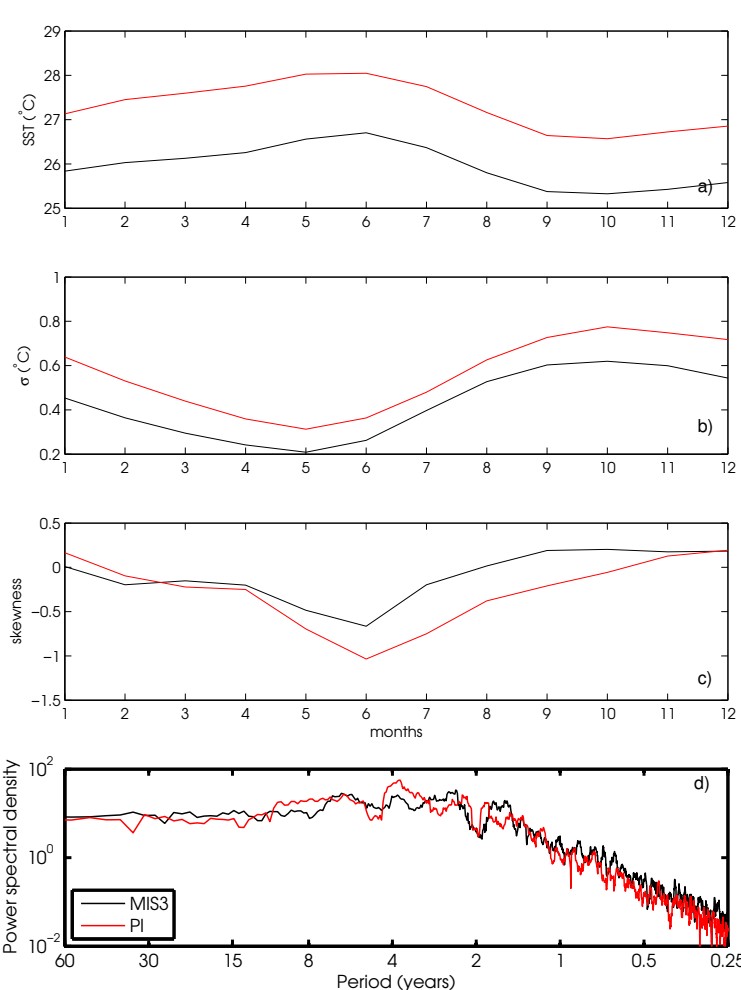

**Figure 15.** Monthly interannual a) SST, b) standard deviation of SST anomalies, c) skewness of SST anomalies in the NINO3.4 region, and d) power spectra of the NINO3.4 index for the MIS3 and PI, respectively.





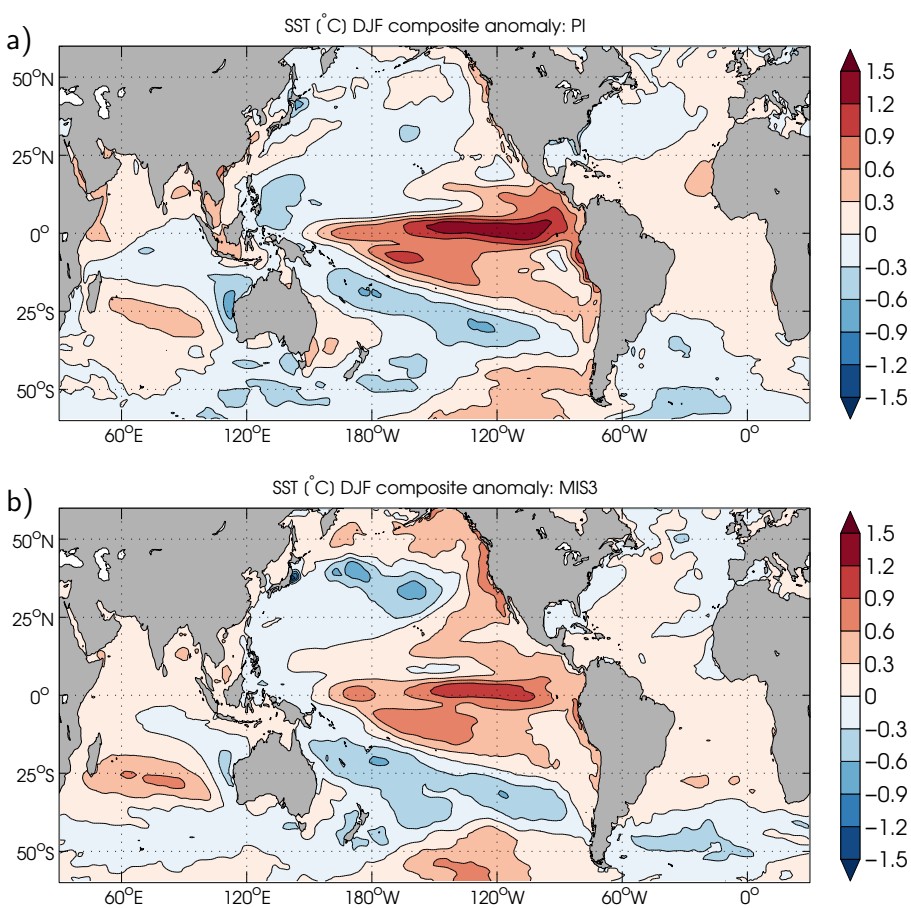

**Figure 16.** Composite DJF SST anomalies during El Niño years for a) PI and b) MIS3.



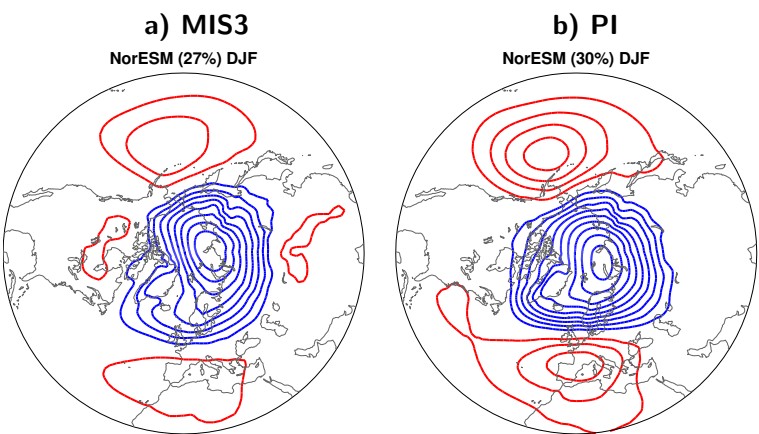

**Figure 17.** a,b) Leading empirical orthogonal function (EOF) of the winter (DJF) mean sea level pressure (SLP) anomalies over the NH (20-90° N) for a) MIS3 and b) PI. The SLP patterns are obtained by regression of anomalies on the leading principal component time series. The contour intervals in both panels are 1 hPa, with the zero line omitted.



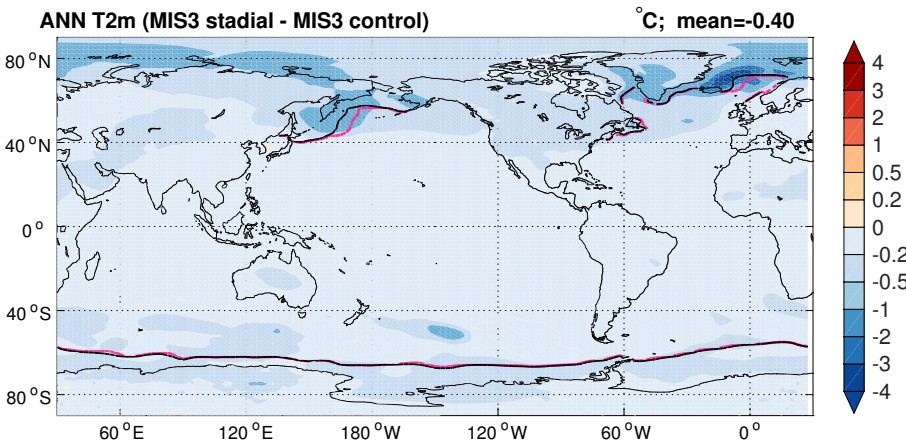

**Figure 18.** Simulated near surface temperature anomaly of MIS3 "stadial" experiment relative to MIS3 interstadial control run (model output averaged between years 2401-2500). The black and magenta lines indicate the 15% March sea ice concentration for the MIS3 interstadial and "stadial" experiments, respectively.



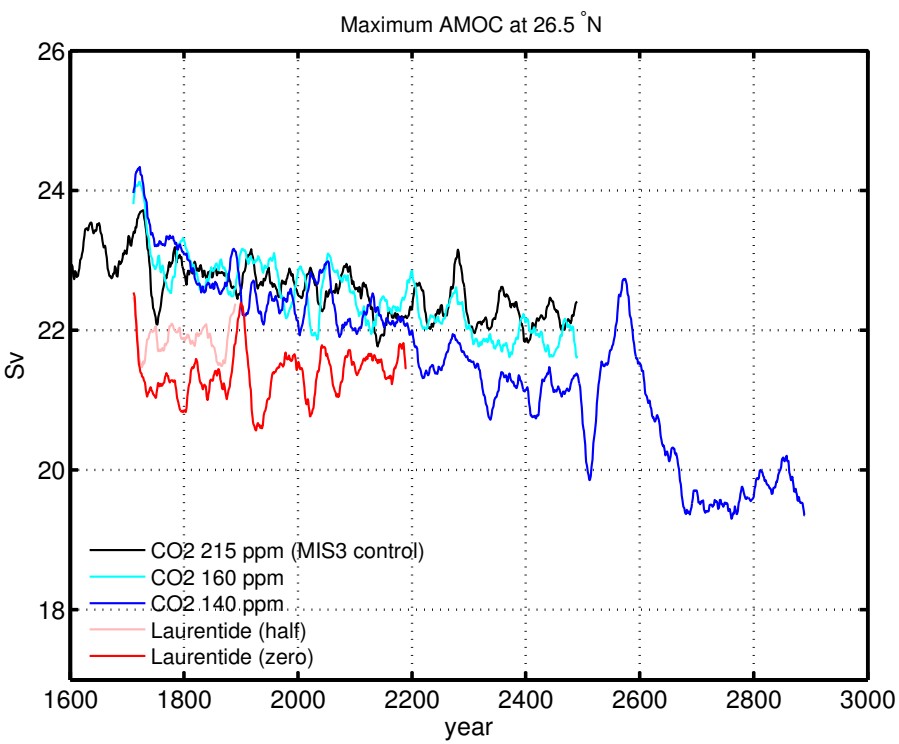

**Figure 19.** Time series of AMOC at 26.5° N for the low $CO_2$ experiments and reduced Laurentide Ice Sheet experiments. The $CO_2$ and ice sheet experiments are branched off from the MIS3 control run from year 1700 and year 2500, respectively, and the latter are shifted forward by 800 years in the figure.





**Table 1.** Forcings and boundary conditions for the MIS3 and PI simulations.

| exp. | MIS3 | PI |
|---|---|---|
| Orbital parameters | | |
|     Eccentricity | 0.013676 | 0.016708 |
|     Obliquity | 23.268° | 23.441° |
|     Perihelion - 180° | 205.94° | 102.72° |
| Trace gases | | |
|     $CO_2$ | 215 ppm | 285 ppm |
|     $CH_4$ | 550 ppb | 792 ppb |
|     $N_2O$ | 260 ppb | 276 ppb |
|     CFC | 0 | 12.5 ppt |
| Solar constant | 1360.9 W m$^{-2}$ | 1360.9 W m$^{-2}$ |
| Ice sheets | data-constrained 38 ka BP | Modern |
| Vegetation | PI + tundra (new land points) | Modern |



**Table 2.** Global mean values for the MIS3 and PI experiments (both averaged between years 1801-2000).

| exp. | MIS3 | PI |
|---|---|---|
| TOA radiation balance | -0.16 W m$^{-2}$ | -0.04 W m$^{-2}$ |
| T2m | 11.6 $^{\circ}$C | 14.5 $^{\circ}$C |
| Precipitation | 2.66 mm day$^{-1}$ | 2.84 mm day$^{-1}$ |
| SST | 17.5 $^{\circ}$C | 18.7 $^{\circ}$C |
| SSS | 34.3 g kg$^{-1}$ | 34.0 g kg$^{-1}$ |
| NINO3.4 $\sigma$ | 0.45 $^{\circ}$C | 0.58 $^{\circ}$C |
| Global ocean T | 1.8 $^{\circ}$C | 3.5 $^{\circ}$C |
| Global ocean S | 35.3 g kg$^{-1}$ | 34.7 g kg$^{-1}$ |
| Ocean Transports | | |
|     Maximum AMOC | 27.5 Sv | 24.3 Sv |
|     AMOC at 26.5$^{\circ}$ N | 22.8 Sv | 20.9 Sv |
|     North Atlantic subtropical gyre | 63.8 Sv | 53.6 Sv |
|     North Atlantic subpolar gyre | 60.7 Sv | 50.3 Sv |
|     Florida Strait | 16.2 Sv | 15.1 Sv |
|     Bering Strait | closed | 1.3 Sv |
|     Barents Sea Opening | 0.5 Sv | 2.8 Sv |
|     Drake Passage | 137.9 Sv | 114.2 Sv |
| Sea ice area | | |
|     NH Mar | 11.1 $\times 10^6$ km$^2$ | 13.5 $\times 10^6$ km$^2$ |
|     NH Sep | 6.3 $\times 10^6$ km$^2$ | 5.2 $\times 10^6$ km$^2$ |
|     SH Mar | 7.3 $\times 10^6$ km$^2$ | 4.4 $\times 10^6$ km$^2$ |
|     SH Sep | 21.5 $\times 10^6$ km$^2$ | 16.4 $\times 10^6$ km$^2$ |