# Peer review of "Equilibrium simulations of Marine Isotope Stage 3 climate"

_Climate of the Past, 2018_

## Referee Comment (RC1) · Anonymous Referee #1 · 13 Dec 2018

Guo and colleagues present an equilibrium simulation of the Marine Isotope Stage 3 (MIS3) with a fully coupled climate model. The simulated climate is very stable and representative of an interstadial climate state with a strong AMOC and relatively high temperatures over Greenland. A stadial climate with a weaker AMOC and lower Greenland temperatures cannot be simulated, not even with typical stadial $CO_2$ concentrations. Sensitivity studies with even lower $CO_2$ concentrations and flat ice sheets support the hypothesis that the NorESM model is very far away from a potential threshold where the climate changes from interstadial to stadial conditions.

The topic of the paper – MIS3 climate state and variability – fits well into the scope of Climate of the Past and is very relevant for the community. There are few fully coupled MIS3 simulations to date and the presented simulations is therefore very valuable as

it adds more data points to the parameter space of glacial forcings and thus helps to understand (1) MIS3 climate variability and (2) the model dependence of glacial climate states.

I recommend the article for publication after some suggested revisions: I believe, the study could be put more into context with existing MIS3 simulations and reconstructions (see general comments below), and a few issues require clarification before publication (see specific comments below).

——————————————— General Comments: ———————————————

The presented MIS3 simulation could be more embedded into the existing literature, both in terms of simulations and existing proxies. Throughout the text, especially in Sect 3.2, the analysis is very descriptive and there are very few comparisons with the existing MIS3 simulations in terms of surface temperature response, sea-ice patterns or AMOC state. The authors mention Barron&Pollard (2002), Van Meerbeeck et al (2009) and Brandefeld et al (2011) in the introduction. It is true that there are not so many coupled simulations with MIS3 boundary conditions, but there are some more MIS3 control simulations available that have been published as reference simulations for hosing experiments, e.g. Xiao Zhang et al (GRL, 2014) or Kawamura et al (Science Advances, 2017, here the information is somewhat hidden in the Supplementary Information).

Often the authors compare their simulations to existing simulations and proxies from the LGM. This is an obvious choice, since there are more simulations and reconstructions available for the LGM than for MIS3. But then these comparisons can be a bit confusing/misleading, since we would not expect the climate state of MIS3 and LGM to be the same. I therefore suggest that the authors go carefully through their manuscript again and check in each case what they want to obtain from the LGM comparison. Can some insight be gained from the LGM /MIS3 differences? If possible it would also be good to have a few more comparisons with existing MIS3 reconstructions, there is

e.g. a recent study by Sessford et al (Paleoceanography, 2018) on water masses and sea-ice in the Denmark strait.

I believe, a more thorough comparison with the existing simulations and MIS3 proxies can help to highlight where the presented MIS3 simulation provides new insight and thus make the study more interesting and relevant.

———————————————— Specific Comments: ————————————————-

p.1, ll.12-15: '[…] questioning the potential for unforced abrupt transitions [...]' In the text you phrase that conclusion quite carefully and refer to the model dependence of MIS3 climate (in)stability. In the abstract, the formulation is perhaps a bit too general, the model dependency should appear here, too.

p.3 - Model description: Would it not be easier to directly describe NorESM1-F rather than describing first how NorESM1-M differs from CCSM4 and then to describe how NorESM1-F differs from NorESM1-M?

p.4, ll.28 – p.5, ll.12: It is not quite clear to me, how the exact MIS3 ice sheets are obtained. Are they assembled from different sources? Why not take them all from the same reconstructions? And why is the Barents Sea so problematic? According to its mean depth it should be open also with 70m lower sea-level, no? Are there conflicting reconstructions?

p.6, ll.20: based on what do you decide that the trend is small? Is there also a threshold value such as for deep ocean salinity in the next sentence?

p.6, ll.30: is the sea-ice drift acceptably small?

p.7, ll.8-12: It is interesting though, that the final simulated MIS3 AMOC is still stronger and deeper than at PI, and the AABW cell is also weaker than at PI, even though AABW is saltier and more ventilated. I'll come back to this issue in a later comment.

p.7, ll.19-22: Are there no MIS3 studies available for the comparisons? (see also

general comments)

p.8, ll.22: The warming in subpolar gyre seems to be more of a dipole. Ist the NAC shift a north-south shift? Can it be seen in the barotropic stream function in Fig. 11?

p.9, ll.22: Why is there more runoff into South China Sea, when precipitation is decreased according to Fig. 8?

p.9/10, AMOC and hydrography section: I find this section somewhat confusing for many reasons.

(1) I think, the LGM comparisons are not very helpful here (see also general comments), as the MIS3 AMOC is expected to be very different from the LGM AMOC. A comparison with the LGM AMOC and hydrography would be more helpful in the discussion, when speculating about reasons for a stable or unstable AMOC. If available, MIS3 comparisons would be more helpful here. From Böhm et al (2015) it should at least be possible to get a qualitative picture of the distribution of northern and southern sourced waters from the eNd measurements.

(2) I am surprised, that the North Atlantic salinity does not increase more than the globla average of 0.6 g/kg. If the mechanism that makes the MIS3 AMOC stronger than the PI AMOC is the same in NorESM than on Muglia&Schmittner (2015) and Klockmann et al (2016/18), I would have expected a much larger salinity increase both at the surface and in the deep North Atlantic.

(3) If more warm NADW is present below 3000m, where does the very cold anomaly in the deep North Atlantic come from? Is that Overflow water then?

(4) Can the anomaly at 500-800 m really be attributed to the Mediterranean Outflow? I would expect the outflow at depths around 1100 m.

(5) What is ideal age? Is it the time since the water mass was in contact with the surface?

(6) If the AABW formation is determined by increased sea-ice formation and brine-rejection, how can it be so well ventilated? I understand from the Ferrari (2014) paper, that AABW was very poorly ventilated because it was upwelled under the ice with little exchange with the atmosphere, and that this is one reason for the glacial $CO_2$ draw-down.

(7) If AABW ventilation and formation increases but the lower overturning cell weakens with less AABW reaching the North Atlantic, where does the AABW go? Is there more AABW in the Pacific?

p.13. ll.11-14: Same as comment (6) above: I understand from the Ferrari (2014) paper, that AABW was very poorly ventilated because it was upwelled under the ice with little exchange with the atmosphere, and that this is one reason for the glacial $CO_2$ draw-down. So how does that fit with a well ventilated AABW?

p.13, ll.27-31: Same as comment (2) above: If the mechanism that makes the MIS3 AMOC stronger than the PI AMOC is the same in NorESM than on Muglia&Schmittner (2015) and Klockmann et al (2016/18), I would have expected a much larger salinity increase both at the surface in the subpolar gyre and in the deep North Atlantic.

p.14, ll.12-13: Xiao Zhang et al (2014) on the other hand find that MIS3 is close to disequilibrium in their simulation.

p.15, ll.1-11: The sensitivity simulations appear quite short, especially the ones with the reduced ice sheets. What determined the length of the simulations? I would argue that the simulations are not in equilibrium, yet. Especially the simulation with 140 ppm could well still be declining – e.g. some of the simulations in Klockmann et al (2018) had about 1500 years of spin up, before the state transitions occurred. I would disagree with the statement that the AMOC strength is unaffected. The responses are small, but both ice sheet reductions and the very low $CO_2$ lead to an AMOC weakening. Whether the weakening is strong enough to produce a stadial climate state is then another question.

Figures: I would say there are already almost too many figures. But still I would like to ask for a figure showing also the deep water formation sites on the Northern hemisphere. I think they could be very helpful for understanding the AMOC stability. I personally would find that more informative than e.g. the insolation in Fig. 1.

———————————————— Technical comments: ————————————————

p.1, ll.8: remove parentheses around 'by ∼13%'

p.1, ll.21-23: reformulate sentence for clarity

p.3, ll.32: Add a reference for HAMOCC

p.5, ll.5 and throughout the text, for the convenience of the reader, take care to distinguish between ice sheets/land ice and sea ice. Sometimes only ice is used.

p.5, ll.10: MSl3 should be MIS3

p.14, ll.24: Remove parentheses around citation

Figure 11: solid contours should indicate negative values and dashed contours positive. They are mixed up in the caption.

Figure 14: This may be a matter of taste; I find it more appropriate to have non diverging colourmaps for the absolute T and S sections in (a) and (c).

References: Xiao Zhang et al (2014), Instability of the Atlantic overturning circulation during Marine Isotope Stage 3, Geophysical Research Letters, https://doi.org/10.1002/2014GL060321

Kawamura et al (2017), State dependence of climatic instability over the past 720,000 years from Antarctic ice cores and climate modeling, Science Advances, DOI: 10.1126/sciadv.1600446 + Supplementary Information

Sessford et al (2018), High-Resolution Benthic Mg/Ca Temperature Record of the Intermediate Water in the Denmark Strait Across D-O Stadial-Interstadial Cycles,

https://doi.org/10.1029/2018PA003370

---

## Referee Comment (RC2) · Anonymous Referee #2 · 10 Jan 2019

This is a study that aims to improve the understanding of the climate of Marine Isotope Stage 3 (MIS3) when the millennial-time scale climate variability occurred most frequently during the last glacial period. The authors perform simulations of MIS3 with a comprehensive state-of-the-art climate model and compare the climate and the oceanic circulation, especially the Atlantic Meridional Overturning Circulation (AMOC) with those of the Preindustrial (PI) and the Last Glacial Maximum (LGM). The authors further show the sensitivity of the MIS3 climate and the AMOC to modifications in the boundary conditions, such as the Laurentide ice sheet and CO2. The model does not exhibit a threshold-type behaviour of the AMOC under low Laurentide ice sheet and low CO2 level. This is interesting, and it offers the community a chance to improve our understanding of the AMOC and discrepancies among models. This paper should

be published because of two reasons. First, it produces important information of the climate of MIS3, which is still rare compared to that of the LGM. Second, it assesses the sensitivity of the MIS3 climate to modifications in the boundary conditions, which is unique compared to previous studies assessing the sensitivity under the PI and the LGM conditions. Nevertheless, I also feel that the important points of this study are still unclear and the manuscript is too long. Below are some suggestions to improve the manuscript.

—General comments—

1. The author should focus more on the results of the sensitivity experiments and perform more analysis in these experiments, since they are the most important and interesting point of this study. In particular, what happens to surface salinity and density over the North Atlantic Deep Water (NADW) formation region and Antarctic bottom water formation (AABW) region when the Laurentide ice sheet and $CO_2$ are modified? Since previous studies have shown that changes in surface salinity and density are very important in understanding the changes in the strength and the threshold of the AMOC (e.g. Montoya and Levermann 2008, Oka et al. 2012, Sun et al. 2016, Buizert and Schmittner 2015, Sherriff-Tadano et al. 2018, Klockmann et al. 2018, Galbraith and de Lavergne 2018), analysis on this point is very important. This analysis will also give very useful information in comparing the results of your model with other climate models. Further analysis on the depth of the AMOC, as well as sea ice cover over the North Atlantic and Southern Ocean should be conducted (e.g. Klockmann et al. 2016, Kawamura et. 2017, Galbraith and de Lavergne 2018).

2. The manuscript is too descriptive and long, which makes the reader difficult to understand the important message of this study. In particular, sections 3.1, 3.2, and 4 are too descriptive. I do understand that these sections show important results, however, they do not give new results and rather follows several previous studies. Unless you compare these simulation results with proxies and other climate models in detail, you should move some part of this section to the Supplementary section. This will help

shorten the manuscript.

—Specific comments—

1 Introduction

The authors should state the significance of this study compared to previous studies more clearly in the last three paragraphs. These points are vague in the manuscript. I understand that the simulation of MIS3 is important since most previous studies conducted simulations of the LGM when they explore the glacial climate. However, in the manuscript, the significance of this study compared to previous MIS3 modelling studies is vague. This point should be clarified.

2 Methods

P4 L30-32: I couldn't quite understand this sentence. Do you just mean that the shape of the ice sheet is prescribed in the model?

P5 L10: MSI3 → MIS3

3 Results

3.1 Model spin-up

I agree that the model has almost reached a quasi-equilibrium state. However, I feel that this section is too long, which makes the reader tired. Please consider reducing the amount of this section. (See also Comment 2).

P7 L11: Where is the location of the open ocean convection over the Southern Ocean in the MIS3 experiment?

3.2 Simulated MIS3 climate

This section is too descriptive and long. Please consider reducing the amount of this section by moving some part of it to the Supplementary. (See also Comment 2).

P7 L33-P8 L1: The strengthening of the surface easterly wind stress over the Irminger

Sea is also caused by the expansion of the Laurentide ice sheet (Sherriff-Tadano et al. 2018).

P8 L30-P9 L3: Please mention that the lowering of CO2 is important in causing the expansion of sea ice and in decreasing the sea surface temperature.

P9 L27: You may remove the first sentence, which is already mentioned in the Introduction.

P9L30-L31: Did you try to say 'The deeper overturning stream function is associated with contracted and weakened AABW'?

P10L21-L25: Kobayashi et al. (2015) also report similar response in their LGM simulation. The decrease in ideal age of the water is attributed to enhanced open ocean convections over the Southern Ocean. You may cite this paper as well.

P11L18-L20: Merkal et al. (2010) also shows similar results in their MIS3 insterstiadial simulation. You may cite this study as well.

P11L32-L33: Is this difference statistically significant?

P12L2-L3: Is this difference statistically significant?

4 MIS3 simulation forced by stadial conditions Please consider reducing the amount of this section by moving some part of it to the Supplementary. (See also Comment 2).

5 Discussion

5.1 Simulated AMOC in MIS3

P13L13: I rather use 'bottom water formation' than 'open ocean convection'.

P13L26-L30: As far as I know, Montoya and Levermann (2008) first showed the potential role of surface winds over the North Atlantic in intensifying the AMOC, Oka et al. (2012) showed that the LGM surface wind enhanced the AMOC with one model, Muglia and Schmittner (2015) confirmed the study of Oka et al. (2012) by performing

analysis with PMIP3 climate models, and Sherriff-Tadano et al. (2018) investigated the processes by which surface winds anomaly induced by the ice sheets enhanced the AMOC. These studies should also be cited in this sentence.

P13L31: Hu et al. (2015) investigated the impact of the closure of Bering Strait on the AMOC. This study should also be cited in this sentence.

5.2 MIS3 sensitivity to CO2 and ice sheet size

Results presented in this section are really interesting! As mentioned in Comment 1, I strongly encourage the authors to perform more analysis on these sensitivity experiments (surface salinity, density and sea ice cover over the NADW and AABW formation region, and the depth of the AMOC). Based on these analysis, you may further discuss the possible cause of differences among previous modelling studies. (Also, if possible, it may be interesting to discuss changes in surface air temperature and precipitation in the half-size Laurentide ice sheet. This analysis can provide an uncertainty of the simulated temperature and precipitation anomalies arising from the uncertainty in the shape of the MIS3 ice sheet. Just a suggestion.)

P15L13: What do you mean by 'ice inhibiting convection'?

Figures

Fig.2: Can you put labels on the contours?

Fig.11: Can you add a figure showing the anomaly? It's difficult to understand the difference between MIS3 and PI from these figures.

References

Buizert, C., & Schmittner, A. (2015). Southern Ocean control of glacial AMOC stability and Dansgaard-Oeschger interstadial duration, Paleoceanography,
30, doi:10.1002/2015PA002795

Hu AX, Meehl GA, Han WQ, Otto-Bliestner B, Abe-Ouchi A, Rosen- bloom N (2015)

Effects of the Bering Strait closure on AMOC and global climate under different background climates. Prog Oceanogr 132:174–196. doi:10.1016/j.pocean.2014.02.004

Kawamura K et al (2017) State dependence of climatic instability over the past 720,000 years from Antarctic ice cores and cli- mate modeling. Sci Adv 3:e1600446

Kobayashi H, Abe-Ouchi A, Oka A (2015) Role of Southern Ocean stratification in glacial atmospheric $CO_2$ reduction evaluated by a three-dimensional ocean general circulation model. Pale- oceanography 30:1202–1216. doi:10.1002/2015PA002786

Merkel U, Prange M, and Schultz M (2010) ENSO variability and teleconnections during glacial climates, Quat. Sci. Rev., 29, 86–100,.

Montoya M, Levermann A (2008) Surface wind-stress threshold for glacial Atlantic overturning. Geophys Res Lett 35:L03608. doi: 10.1029/2007gl032560

Oka A, Hasumi H, Abe-Ouchi A (2012) The thermal threshold of the Atlantic meridional overturning circulation and its control by wind stress forcing during glacial climate. Geophys Res Lett 39:L09709. doi:10.1029/2007gl029475

Sherriff-Tadano, S., Abe-Ouchi, A., Yoshimori, M., Oka, A., & Chan, W.-L. (2018). Influence of glacial ice sheets on the Atlantic meridional overturning circulation through surface wind change. Climate Dynamics, 50, 2881-2903. doi:10.1007/s00382-017-3780-0

Sun, S., Eisenman, I., & Stewart A. (2016). The influence of Southern Ocean surface buoyancy forcing on glacial-interglacial changes in the global deep ocean stratification, Geophysical Research Letters, 43, 8124–8132. doi:10.1002/2016GL070058

---

## Referee Comment (RC3) · Anonymous Referee #3 · 14 Jan 2019

In the manuscript cp-2018-165 entitled "Equilibrium simulations of Marine Isotope Stage 3 climate" by Guo et al., the authors compared the simulated climate mean state of Marine Isotope Stage 3 (MIS3) and preindustrial (PI) era using the Norwegian Earth System Model (NorESM). They found a cooler climate in MIS3 relative to PI conditions with a thicker and more expanded sea ice. The AMOC strengthen by 13% with reduced AABW reaching the North Atlantic. Moreover the AABW production actually increases due to the increased sea ice cover in the southern oceans in association to the cooler MIS3 climate. They also show a reduced ENSO and NAM variability. Finally, by doing a few sensitivity simulations by reducing CO2 concentration or ice sheet height in the North America, they suggest that abrupt transitions of climate from interstadial to stadial state is not likely, and raised the question whether abrupt climate transition would

be possible without changes of external forcings. I found this manuscript is well written and easy to follow. The results are interesting to the readership of the Climate Past community. Thus I would like to recommend this manuscript to be accepted after some revision:

Comments:

1. The simulations are primarily focus on the PI and MIS3 climate background, thus it is not surprising that the climate states are stable for both conditions. One question the authors did not specifically clearly state is the initial condition of these runs. It seems that both PI and MIS3 runs state from the same ocean initial state, except an increase of the mean salinity for MIS3 run. Is this true? If so, how will this affect the model sensitivity when icesheet height or CO2 concentration changes?

2. The authors tested the model response of CO2 reduction be 15 ppmv. The question here is whether the MIS3 stadial climate is caused by CO2 reduction or by changes of the AMOC? It seems that the authors assumed that the CO2 reduction is the cause, is it true?

3. Although the experiments done by the authors don't show significant AMOC changes, it seems this is not enough to question the possible multi-equilibria of AMOC, especially the experimental design may not serve the purpose of the authors. A better test is to check whether AMOC has multi-equilibria in the NorESM under glacial condition. If yes, then the authors can test whether an abrupt transition of the AMOC is possible with the absence of the external forcing change. It may be important to test the small changes of the external forcings and whether this small changes can bring the climate state to a critical point in which even smaller changes in freshwater forcing is capable to collapse the AMOC.

---

## Referee Comment (RC4) · Anonymous Referee #4 · 16 Jan 2019

The manuscript "Equilibrium simulations of Marine Isotope Stage 3 Climate" by Guo and colleagues well present a new MIS3 simulation in their model NorESM1-F. They employed a latest (new) ice sheet configuration without a large Fennoscandian ice sheet to conduct their MIS3-38ka simulation. By several attempts exploring the tipping point/bifurcation in their 38ka simulation, the authors claim that their 38ka simulation in NorESM1-F is too stable to reach a tipping point that was commonly used to explain the millennial-scale variability during the MIS3 in previous modeling studies.

The authors first described their 38ka simulation results in very detail (although it can be more compact), in accompany with a comparison with previous LGM simulations. The LGM is a good reference (LGM) to compare with, but there is lack of detailed discussion of their differences.  The 38ka simulation was integrated for 2500 model

years. The author argue that their simulation is almost in a quasi-equilibrium since the salinity trend in the Atlantic is less than 0.06 g/kg. How is it defined because it remains possible that the deep ocean is not in a quasi-equilibrium state if there exists a robust salinity increase in the AABW formation region. This is also my concern for their sensitivity simulation with lower CO2 levels. It is very likely that polar regions need a much longer time scale (> 1000 model years) to cool down, producing the cold enough bottom/deep water masses and so the glacial ocean structure and circulation.

The authors also assess the simulated climate mode variability in their simulations. It makes the manuscript more comprehensive. However, I do not find a clear connection to the following investigation of the AMOC bistability? why does the "stadial" condition under 40ka boundary not include freshwater forcing in the North Atlantic? I fully understand the authors' purpose, but since H4 did feature a robust freshwater input, it would be more comparable and reasonable to force a stadial climate with the North Atlantic freshwater forcing under 40ka-38ka boundary conditions. If the authors would like to investigate the changes in climate variability in stadial conditions, I would suggest conducting the hosing under 38ka boundary condition. This will largely reduce the difficulty in the discussion of differences between interstadial and stadial runs. The present "stadial" climate can be included in the section regarding exploration of AMOC bistability.

In the discussion part, the author design several sensitivity experiments to explore the potential nonlinear behavior of their 38ka climate. The experiments are reasonable and clear, which can provide end-members of climate responses to glacial-interglacial variations regarding ice volume and pCO2. It is a promising try here although probably these runs (especially lower CO2 runs) are not in quasi-equilibrium, therefore the runs are not as conclusive as the authors argued. In addition, one conceptual mistake in the manuscript is that spontaneous oscillation does not share the same definition with the AMOC bistability, but rather a hopf bifurcation feature. The authors shall go through the manuscript carefully to distinguish their difference.

Overall I find the manuscript is interesting and well within the scope of Clim. Past. It can be accepted for publication after some modifications in the structure as well as refinements of model results description and discussion.
* * *

---

## Author Comment (AC1) · 26 Feb 2019

Response to Reviewer #1:

Please see response to the referee's comments in blue font.

Guo and colleagues present an equilibrium simulation of the Marine Isotope Stage 3 (MIS3) with a fully coupled climate model. The simulated climate is very stable and representative of an interstadial climate state with a strong AMOC and relatively high temperatures over Greenland. A stadial climate with a weaker AMOC and lower Greenland temperatures cannot be simulated, not even with typical stadial CO2 concentrations. Sensitivity studies with even lower C02 concentrations and flat ice sheets support the hypothesis that the NorESM model is very far away from a potential threshold where the climate changes from interstadial to stadial conditions.

The topic of the paper – MIS3 climate state and variability – fits well into the scope of Climate of the Past and is very relevant for the community. There are few fully coupled MIS3 simulations to date and the presented simulations is therefore very valuable as it adds more data points to the parameter space of glacial forcings and thus helps to understand (1) MIS3 climate variability and (2) the model dependence of glacial climate states.

I recommend the article for publication after some suggested revisions: I believe, the study could be put more into context with existing MIS3 simulations and reconstructions (see general comments below), and a few issues require clarification before publication (see specific comments below).

We thank the reviewer for his/her thorough assessment and constructive comments on our manuscript. We respond to the reviewer's comments below point by point.

———————————————— General Comments: ————————————————

The presented MIS3 simulation could be more embedded into the existing literature, both in terms of simulations and existing proxies. Throughout the text, especially in Sect 3.2, the analysis is very descriptive and there are very few comparisons with the existing MIS3 simulations in terms of surface temperature response, sea-ice patterns or AMOC state. The authors mention Barron&Pollard (2002), Van Meerbeeck et al (2009) and Brandefeld et al (2011) in the introduction. It is true that there are not so many coupled simulations with MIS3 boundary conditions, but there are some more MIS3 control simulations available that have been published as reference simulations for hosing experiments, e.g. Xiao Zhang et al (GRL, 2014) or Kawamura et al (Science Advances, 2017, here the information is somewhat hidden in the Supplementary Information).

Often the authors compare their simulations to existing simulations and proxies from the LGM. This is an obvious choice, since there are more simulations and reconstructions available for the LGM than for MIS3. But then these comparisons can be a bit confusing/misleading, since we would not expect the climate state of MIS3 and LGM to be the same. I therefore suggest that the authors go carefully through their manuscript again and check in each case what they want to obtain from the LGM comparison. Can some insight be gained from the LGM /MIS3 differences? If possible it would also be good to have a few more comparisons with existing MIS3 reconstructions, there is e.g. a recent study by Sessford et al (Paleoceanography, 2018) on water masses and sea-ice in the Denmark strait.

I believe, a more thorough comparison with the existing simulations and MIS3 proxies can help to highlight where the presented MIS3 simulation provides new insight and thus make the study more interesting and relevant.

> Thanks for the insightful comments and very relevant references. Following the reviewer's suggestion, we have expanded the discussion with a more in-depth comparison of our results with published MIS3 simulations and proxy records; these include the modelling studies of e.g. Merkel et al., 2010; Xiao Zhang et al., 2014; Kawamura et al., 2017,

and proxy studies of, e.g. Böhm et al. 2015; Sessford et al., 2018. There are certainly more MIS3 proxy based studies, however we choose to focus on comparison to the Nordic Seas region.

As for the reviewer's comment on comparison with previous LGM simulations, we do agree that the two periods are distinct, e.g. MIS3 features reduced ice volume and a relatively warm climate compared to the LGM. We have carefully revised the manuscript, and removed the discussion on the comparison of SST and AMOC strength with LGM PMIP studies. We retained the comparison of near surface temperature - considering the close proximity of the two periods in time, and the much larger number of LGM proxy/modelling studies, we think it is useful to make the comparison (e.g. Van Meerbeeck et al. 2009), with caveats in mind of course. We've therefore added the following to the updated manuscript: "Compared to the amount of MIS3 studies, there is a rich literature on both the simulation and reconstruction of the LGM climate. With both similarities as well as apparent differences with regard to the external forcing and the climate, it can be useful to compare the climate of the two periods..."

————————————————————Specific Comments:—————————————————-
p.1, ll.12-15: '[. . .] questioning the potential for unforced abrupt transitions [...]' In the text you phrase that conclusion quite carefully and refer to the model dependence of MIS3 climate (in)stability. In the abstract, the formulation is perhaps a bit too general, the model dependency should appear here, too.

> Following the reviewer's comment, we have edited the abstract making it consistent with the conclusion and including the reference to model dependency.

p.3 - Model description: Would it not be easier to directly describe NorESM1-F rather than describing first how NorESM1-M differs from CCSM4 and then to describe how NorESM1-F differs from NorESM1-M?

> NorESM family of models is based on CCSM4, and NorESM1-M is the first documented version of the family. NorESM1-F is developed based on the NorESM1-M version. To clarify, we've rewritten the first sentence in Section 2.1 as "The NorESM family is based on the Community Climate System Model version 4..."

p.4, ll.28 – p.5, ll.12: It is not quite clear to me, how the exact MIS3 ice sheets are obtained. Are they assembled from different sources? Why not take them all from the same reconstructions? And why is the Barents Sea so problematic? According to its mean depth it should be open also with 70m lower sea-level, no? Are there conflicting reconstructions?

> The global ice sheet data used in our simulation was kindly provided by Lev Tarasov; the data documenting the individual ice sheets were either published separately (as cited in the manuscript) or unpublished (e.g. the Eurasian ice sheet). Regarding Barents Sea, intuitively it should be open with a 70 m reduction of sea level; however, it is likely that there could be grounded ice that actually closes the sea - geological evidence on the existence of grounded ice is sparse and not firm though, as mentioned in the manuscript. Given that the reconstructions are uncertain and the potential impact is significant, we have chosen to discuss this in more detail in the manuscript.

p.6, ll.20: based on what do you decide that the trend is small? Is there also a threshold value such as for deep ocean salinity in the next sentence?
p.6, ll.30: is the sea-ice drift acceptably small?

> There is not any 'hard' threshold to evaluate the model drift, as far as we are aware of; this is true for both modern as well as paleoclimate simulations. We believe that the trends of the evaluated metrics are acceptably small, although we do think that a longer integration would certainly be an advantage. Unfortunately, we are limited by computational time to run the model longer. However, in the revised manuscript we have removed the sentence "For the NorESM MIS3

simulation, we deem the aforementioned global mean ocean cooling trend to be small."

p.7, ll.8-12: It is interesting though, that the final simulated MIS3 AMOC is still stronger and deeper than at PI, and the AABW cell is also weaker than at PI, even though AABW is saltier and more ventilated. I'll come back to this issue in a later comment.

> We respond to the reviewer's comment regarding AMOC below.

p.7, ll.19-22: Are there no MIS3 studies available for the comparisons? (see also general comments)

> There are indeed MIS3 studies available for comparison. As discussed above, we have added the following to the revised manuscript: "For comparison, The CCSM3 MIS3 simulation (with 35 ka boundary conditions) by Merkel et al. (2010) reported a cooling of 3.4 ◦C, whereas using a different version of CCSM3 configured with 38 ka boundary conditions, Zhang et al. (2014b) reported a cooling of 3.5 ◦C. A stadial simulation with 44 ka boundary conditions (Brandefelt et al., 2011) shows a much cooler climate, e.g., 5.5 ◦C compared to the recent past."

p.8, ll.22: The warming in subpolar gyre seems to be more of a dipole. Ist the NAC shift a north-south shift? Can it be seen in the barotropic stream function in Fig. 11?

> The NAC shifts during MIS3 compared to that in PI can be seen from the sea level anomaly map shown below (MIS3 minus PI). The shift is not apparent from the barotropic stream function which is the vertical integration of volume transport for the whole water column, whereas NAC is located in the upper few hundred meters.

[Figure]

p.9, ll.22: Why is there more runoff into South China Sea, when precipitation is decreased according to Fig. 8?

> This is mainly because a new river routing map is generated in our MIS3 configuration due to the change of land-sea mask, leading to large catchments and several arguably 'artificial' rivers flowing into the South China Sea. We've added the new river routing map in the supplementary material, and have also updated the original text accordingly as "The fresh surface water in the South China Sea during MIS3 is due to increased runoff, that is related to the newly generated river routing in this region owing to the change of land/sea mask."

p.9/10, AMOC and hydrography section: I find this section somewhat confusing for many reasons.
(1) I think, the LGM comparisons are not very helpful here (see also general comments), as the MIS3 AMOC is expected to be very different from the LGM AMOC. A comparison with the LGM AMOC and hydrography would be more helpful in the discussion, when speculating about reasons for a stable or unstable AMOC. If available, MIS3 comparisons would be more helpful here. From Böhm et al (2015) it should at least be possible to get a qualitative picture of the distribution of northern and southern sourced waters from the eNd measurements.

> Following the reviewer's comment, we have removed the LGM comparison here (see also our response to the reviewer's general comments). Instead we focus on comparison to MIS3.

We have added the following MIS3 reference to the revised manuscript: "Zhang et al. (2014b) reported a similar strengthening of AMOC during MIS3 (38 ka boundary conditions), e.g., 15.4 Sv which is 1.5 Sv stronger than their PI control simulation, and is much weaker than our simulated strength of AMOC at MIS3. Zhang et al. (2014b) also simulated a shallower upper cell of AMOC, in contrast to the NorESM simulation."

We have also included a comparison with Böhm et al (2015): "The AMOC during interstadials is accompanied by active deep water formation in the North Atlantic, with persistent contributions from the northern sourced water."

(2) I am surprised, that the North Atlantic salinity does not increase more than the globla average of 0.6 g/kg. If the mechanism that makes the MIS3 AMOC stronger than the PI AMOC is the same in NorESM than on Muglia&Schmittner (2015) and Klockmann et al (2016/18), I would have expected a much larger salinity increase both at the surface and in the deep North Atlantic.

> Intuitively, one would expect a larger increase of Atlantic salinity, given a stronger AMOC and closure of the Bering Strait. However, it is possible that the MIS3 circulation can move salt across basins, for example from the Atlantic to the Pacific, causing the salinity increase in the North Atlantic < 0.6 g/kg. One possibility is that the enhanced AMOC leads to a larger salt export at the southern basin boundary, if the change in surface salt import from Indian Ocean does not fully compensate for the change in NADW related export. Klockmann et al. (2016) also showed less increase of salinity (relative to the global addition of salt due to sea level lowering) in the North Atlantic although with a stronger AMOC at LGM (c.f. their Fig. 5f). While the effect of closing the Bering Strait is clearly visible in the spatial SSS changes (Fig. 9) – showing e.g. a freshening in the Bering Sea and tendency to more saline Canadian Arctic and western North Atlantic – it is not unlikely our model underestimates this effect as a consequence of the model's routing of freshwater on land. The river routing map (added to the supplementary material) shows large catchments for the Canadian Arctic and North Atlantic and only a small catchment for northeast Pacific. Likely the lack of Atlantic salinity response can be mitigated by manually correcting/tuning the routing map (e.g. based on geological evidence) to route more freshwater into the North Pacific instead of Arctic, something we will consider in future studies. Another potential factor may be changes in the moisture transport across Central America. Given the size of the current paper, we think a more detailed

investigation of the hydrological cycle covering the above aspects should be better done elsewhere.

(3) If more warm NADW is present below 3000m, where does the very cold anomaly in the deep North Atlantic come from? Is that Overflow water then?

> First of all, the global ocean is colder during MIS3 than PI (e.g. 1.7 deg C). Also, AABW, although reduced in volume in the deep Atlantic, is colder and contributes to the cold anomaly therein.

It is a possibility that overflow water contributes to the cold anomaly in the deep North Atlantic since large parts of the Nordic Seas are not ice covered in the MIS3 simulation allowing dense water formation in high latitudes to continue.

(4) Can the anomaly at 500-800 m really be attributed to the Mediterranean Outflow? I would expect the outflow at depths around 1100 m.

> yes; the figure below shows a cross-section from the western Atlantic to the Mediterranean along the same latitude: left column - PI; right column - MIS3; top row - temperature; bottom row - salinity. One can see that the change of Mediterranean outflow during MIS3 contributes to the temperature/salinity anomalies at 500-800 m.

[Figure]

A weaker signature of Mediterranean outflow in the MIS3 simulation is plausible as evaporation over the Mediterranean basin is expected to be

reduced under the colder climate i.e. less outflow/inflow is necessary to maintain the freshwater balance of the Mediterranean Sea.

(5) What is ideal age? Is it the time since the water mass was in contact with the surface?

> Yes - it is the time since the water mass last made contact with the surface. We have included this in the revised manuscript.

(6) If the AABW formation is determined by increased sea-ice formation and brine- rejection, how can it be so well ventilated? I understand from the Ferrari (2014) paper, that AABW was very poorly ventilated because it was upwelled under the ice with little exchange with the atmosphere, and that this is one reason for the glacial CO2 draw- down.

> The reviewer is right. The increased sea ice formation and the associated brine rejection contribute to the formation and salinitification of AABW, but one cannot directly link it to enhanced ventilation.

The model does show an enhanced ventilation in the Southern Ocean during MIS3, e.g., the figure below shows the zonal mean ideal age in the Southern Ocean for the MIS3 and PI experiments. It is evident from the figure that AABW is more ventilated during MIS3 compared to PI. The strengthening of ventilation can be caused by processes such as changes in air-sea heat/salt flux, wind etc.

We've therefore modified the text accordingly, and included the figure below into the supplementary material.

[Figure]

[Figure]

(7) If AABW ventilation and formation increases but the lower overturning cell weakens with less AABW reaching the North Atlantic, where does the AABW go? Is there more AABW in the Pacific?

> Yes. The lower overturning cell associated with AABW in the Atlantic is weakened, but globally it is strengthened - see below the global MOC in isopycnal space for MIS3 (left panel below) and PI (right panel below).

[Figure]

[Figure]

In the Pacific and Indian Ocean, it is also clear that MIS3 (left panel below) features a stronger deep cell associated with AABW compared to that in PI (right panel below).

[Figure]

[Figure]

p.13. ll.11-14: Same as comment (6) above: I understand from the Ferrari (2014) paper, that AABW was very poorly ventilated because it was upwelled under the ice with little exchange with the atmosphere, and that this is one reason for the glacial $CO_2$ draw-down. So how does that fit with a well ventilated AABW?

> Please see our response to comment (6) above.

p.13, ll.27-31: Same as comment (2) above: If the mechanism that makes the MIS3 AMOC stronger than the PI AMOC is the same in NorESM than on Muglia&Schmittner (2015) and Klockmann et al (2016/18), I would have expected a much larger salinity increase both at the surface in the subpolar gyre and in the deep North Atlantic.

> Please see our response to comment (2) above.

p.14, ll.12-13: Xiao Zhang et al (2014) on the other hand find that MIS3 is close to disequilibrium in their simulation.

> Yes, we agree that the MIS3 simulation by Xiao Zhang et al (2014) is close to disequilibrium, although this study explores the AMOC instability with the freshwater approach. We have added the following into the updated text: "In addition, with freshwater flux as the external forcing, the MIS3 simulation reported by Zhang et al. (2014b) is close to disequilibrium and model bi-stability."

p.15, ll.1-11: The sensitivity simulations appear quite short, especially the ones with the reduced ice sheets. What determined the length of the simulations? I would argue that the simulations are not in equilibrium, yet. Especially the simulation with 140 ppm could well still be declining – e.g. some of the simulations in Klockmann et al (2018) had about 1500 years of spin up, before the state transitions occurred. I would disagree with the statement that the AMOC strength is unaffected. The responses are small, but both ice sheet reductions and the very low CO2 lead to an AMOC weakening. Whether the weakening is strong enough to produce a stadial climate state is then another question.

> We agree with the reviewer that the sensitivity experiments are not long enough for quasi-equilibrium states. Computing resource is one concern. We terminated the sensitivity experiments with reduced ice sheet, as we did not see any sign/trend of further AMOC reduction and growth of sea ice for these two experiments. We also agree that the 140 ppm experiment, still with a declining trend, has the potential to reach a weak mode of AMOC. However, given that 140 ppm is already a very low level (e.g. compared to that in Zhang et al., 2017 & Klockmann et al., 2018) and that the sensitivity experiment has run for > 1000 years, it indicates that our simulated MIS3 climate is far away from the bifurcation point. We have added the following to the discussion: "…, despite that the simulations are limited in length (200 to 1000 years) and we therefore cannot fully exclude that further equilibration could bring the climate into a more unstable regime."

It is true that we should not claim that the strength of AMOC is not affected in the sensitivity experiments. We have modified this in the updated manuscript, e.g. "... AMOC is only slightly reduced" for the ice sheet sensitivity experiments, whereas for the CO2 sensitivity experiments, "the AMOC, although weakened by several Sv, still remains strong".

Figures: I would say there are already almost too many figures. But still I would like to ask for a figure showing also the deep water formation sites on the Northern hemisphere. I think they could be very helpful for understanding the AMOC stability. I personally would find that more informative than e.g. the insolation in Fig. 1.

> Following the reviewer (and another reviewer)' comment, we have moved some text/figures to the supplementary material to enhance the readability of the manuscript. We have also added a map showing both the MIS3 and PI mixed layer depth (an indication of deep water formation region) into the supplementary material.

———————————————— Technical comments:————————————————

p.1, ll.8: remove parentheses around 'by ~13%'

> removed.

p.1, ll.21-23: reformulate sentence for clarity

> We have rephrased the sentence as "Correlated with the rapid warming of Greenland temperature (up to 15 $^{\circ}$C within a few decades during the stadial-to-interstadial transition), the North Atlantic and Nordic Seas are subject to abrupt climate transitions as interpreted from a number of marine sediment cores...".

p.3, ll.32: Add a reference for HAMOCC

> added, e.g. Maier-Reimer (1993), Maier-Reimer et al. (2005)

p.5, ll.5 and throughout the text, for the convenience of the reader, take care to distinguish between ice sheets/land ice and sea ice. Sometimes only ice is used.

> We've searched through the manuscript to make sure that different types of ice are distinguished.

p.5, ll.10: MSI3 should be MIS3

> corrected.

p.14, ll.24: Remove parentheses around citation

> removed.

Figure 11: solid contours should indicate negative values and dashed contours positive. They are mixed up in the caption.

> revised.

Figure 14: This may be a matter of taste; I find it more appropriate to have non diverging colourmaps for the absolute T and S sections in (a) and (c).

> We would prefer to stay with the current colour maps in this figure, as we would like to have a consistent colour map with previous NorESM evaluations, e.g., Bentsen et al. (2013), Guo et al. (2019).

References: Xiao Zhang et al (2014), Instability of the Atlantic overturning circulation during Marine Isotope Stage 3, Geophysical Research Letters, https://doi.org/10.1002/2014GL060321
Kawamura et al (2017), State dependence of climatic instability over the past 720,000 years from Antarctic ice cores and climate modeling, Science Advances, DOI: 10.1126/sciadv.1600446 + Supplementary Information

Sessford et al (2018), High-Resolution Benthic Mg/Ca Temperature Record of the Intermediate Water in the Denmark Strait Across D-O Stadial-Interstadial Cycles,https://doi.org/10.1029/2018PA003370

---

## Author Comment (AC2) · 26 Feb 2019

Response to Reviewer #2:

We respond to the referee's comments in blue font below.

This is a study that aims to improve the understanding of the climate of Marine Isotope Stage 3 (MIS3) when the millennial-time scale climate variability occurred most frequently during the last glacial period. The authors perform simulations of MIS3 with a comprehensive state-of-the-art climate model and compare the climate and the oceanic circulation, especially the Atlantic Meridional Overturning Circulation (AMOC) with those of the Preindustrial (PI) and the Last Glacial Maximum (LGM). The authors further show the sensitivity of the MIS3 climate and the AMOC to modifications in the boundary conditions, such as the Laurentide ice sheet and CO2. The model does not exhibit a threshold-type behaviour of the AMOC under low Laurentide ice sheet and low CO2 level. This is interesting, and it offers the community a chance to improve our understanding of the AMOC and discrepancies among models. This paper should be published because of two reasons. First, it produces important information of the climate of MIS3, which is still rare compared to that of the LGM. Second, it assesses the sensitivity of the MIS3 climate to modifications in the boundary conditions, which is unique compared to previous studies assessing the sensitivity under the PI and the LGM conditions. Nevertheless, I also feel that the important points of this study are still unclear and the manuscript is too long. Below are some suggestions to improve the manuscript.

We thank the reviewer for his/her thorough assessment and constructive comments on our manuscript. We respond to the reviewer's general and specific comments below point by point.

—General comments—
1. The author should focus more on the results of the sensitivity experiments and perform more analysis in these experiments, since they are the most important and interesting point of this study. In particular, what happens to surface salinity and density over the North Atlantic Deep Water (NADW) formation region and Antarctic bottom water

formation (AABW) region when the Laurentide ice sheet and CO2 are modified? Since previous studies have shown that changes in surface salinity and density are very important in understanding the changes in the strength and the threshold of the AMOC (e.g. Montoya and Levermann 2008, Oka et al. 2012, Sun et al. 2016, Buizert and Schmittner 2015, Sherriff-Tadano et al. 2018, Klockmann et al. 2018, Galbraith and de Lavergne 2018), analysis on this point is very important. This analysis will also give very useful information in comparing the results of your model with other climate models. Further analysis on the depth of the AMOC, as well as sea ice cover over the North Atlantic and Southern Ocean should be conducted (e.g. Klockmann et al. 2016, Kawamura et. 2017, Galbraith and de Lavergne 2018).

> We agree with the reviewer that the sensitivity experiments related to external forcing should be of high interest to many. However, in this work, we would prefer not to weigh the sensitivity experiments over the results of the MIS3 control simulation. There is a rich literature on LGM simulations, whereas there are few MIS3 simulations with a state-of-the-art climate model. Besides, most of the previous studies on AMOC bi-stability/abrupt climate change in the last glacial are configured with boundary conditions of either LGM or PI, thereby with a deviation/bias already in the control experiment. We therefore believe that a comprehensive assessment of the simulated MIS3 climate would be necessary and could serve as a useful reference and basis for dedicated, future MIS3 simulation studies with the same model that focus on climate sensitivity. However, we agree with the reviewer that the original manuscript can be more compact, and we have accordingly moved certain results to the supplementary material (see our response to the next comment).

In addition, in response to the reviewer's suggestion on more detailed analysis of the sensitivity experiments, we have added a section in the supplementary material showing the response of SSS, winter sea ice, and AMOC in depth-latitude space. As we have mentioned in the first draft of manuscript, NorESM is in a relatively stable state and stays far away from the threshold for state transitions; as a consequence, the

response of the climate system in the sensitivity experiments are relatively small, as reflected in the changes of metrics mentioned above. Specifically, as the changes in e.g. SSS, sea ice, and AMOC geometry are highly related to the strength of AMOC, which is only weakly reduced in the sensitivity experiments; therefore, significant changes in SSS, sea ice, and AMOC geometry etc. would not be expected. We have also added some discussions in the manuscript.

2. The manuscript is too descriptive and long, which makes the reader difficult to understand the important message of this study. In particular, sections 3.1, 3.2, and 4 are too descriptive. I do understand that these sections show important results, however, they do not give new results and rather follows several previous studies. Unless you compare these simulation results with proxies and other climate models in detail, you should move some part of this section to the Supplementary section. This will help shorten the manuscript.

> As also mentioned in our response to the previous comment, we have moved quite a bit of results in sections 3.1, 3.2, and 4 to the supplementary material; these include the time series of sea ice, several atmospheric diagnosis, the whole section of "Modes of variability", and the discussion on the "stadial" experiment. We believe that the results are presented in a more succinct manner in the updated manuscript.

—Specific comments—
1 Introduction
The authors should state the significance of this study compared to previous studies more clearly in the last three paragraphs. These points are vague in the manuscript. I understand that the simulation of MIS3 is important since most previous studies conducted simulations of the LGM when they explore the glacial climate. However, in the manuscript, the significance of this study compared to previous MIS3 modelling studies is vague. This point should be clarified.

> Following the reviewer' comment, we have rephrased the text in Introduction to highlight the significance of our study.

**2 Methods**

P4 L30-32: I couldn't quite understand this sentence. Do you just mean that the shape of the ice sheet is prescribed in the model?

> Yes, exactly; we have clarified this in the updated manuscript: "NorESM1-F does not have a dynamic land ice component, and the assumed ice sheet extent and elevation during MIS3 compared to present day are prescribed."

P5 L10: MSI3 → MIS3 3 Results

> corrected.

**3.1 Model spin-up**

I agree that the model has almost reached a quasi-equilibrium state. However, I feel that this section is too long, which makes the reader tired. Please consider reducing the amount of this section. (See also Comment 2).

> Following the reviewer' suggestion, we have moved the text/discussion on sea ice to the supplementary material. We think that the rest of metrics are important for the evaluation of model drift, and therefore would prefer to keep them.

P7 L11: Where is the location of the open ocean convection over the Southern Ocean in the MIS3 experiment?

> In the Weddell Sea region and the Pacific and Indian sectors of the Southern Ocean. We have included a figure of austral winter mixed layer depth in the supplementary material.

**3.2 Simulated MIS3 climate**

This section is too descriptive and long. Please consider reducing the amount of this section by moving some part of it to the Supplementary. (See also Comment 2).

> We have done so. Please see our response to the reviewer's general comment 2.

P7 L33-P8 L1: The strengthening of the surface easterly wind stress over the Irminger Sea is also caused by the expansion of the Laurentide ice sheet (Sherriff-Tadano et al. 2018).

> yes, good point; reference cited.

P8 L30-P9 L3: Please mention that the lowering of CO2 is important in causing the expansion of sea ice and in decreasing the sea surface temperature.

> We have added the following to the beginning of the subsection: "The reduced level of $CO_2$ during MIS3 is important in lowering SST and in causing the expansion of sea ice."

P9 L27: You may remove the first sentence, which is already mentioned in the Introduction.

> sentence removed.

P9L30-L31: Did you try to say 'The deeper overturning stream function is associated with contracted and weakened AABW'?

> The original statement is a bit misleading; we have rephrased it as "The lower overturning cell associated with AABW is contracted and weakened."

P10L21-L25: Kobayashi et al. (2015) also report similar response in their LGM simulation. The decrease in ideal age of the water is attributed to enhanced open ocean convections over the Southern Ocean. You may cite this paper as well.

> We thank the reviewer for the suggestion of this study. We have referred to this paper in the revised manuscript: "Kobayashi et al. (2015)

reported similar response of LGM water mass age in the Southern Ocean owing to enhanced open ocean convections."

P11L18-L20: Merkal et al. (2010) also shows similar results in their MIS3 insterstiadial simulation. You may cite this study as well.
P11L32-L33: Is this difference statistically significant?
P12L2-L3: Is this difference statistically significant?

> The study by Merkel et al. (2010) is very relevant and has been cited. Following the reviewer's second general comment, we have moved the section of ENSO/NAM to the supplementary material.

4 MIS3 simulation forced by stadial conditions Please consider reducing the amount of this section by moving some part of it to the Supplementary. (See also Comment 2).

> We have done so. Please see our response to the reviewer's general comment 2.

5 Discussion
5.1 Simulated AMOC in MIS3
P13L13: I rather use 'bottom water formation' than 'open ocean convection'.

> modified.

P13L26-L30: As far as I know, Montoya and Levermann (2008) first showed the potential role of surface winds over the North Atlantic in intensifying the AMOC, Oka et al. (2012) showed that the LGM surface wind enhanced the AMOC with one model, Muglia and Schmittner (2015) confirmed the study of Oka et al. (2012) by performing analysis with PMIP3 climate models, and Sherriff-Tadano et al. (2018) investigated the processes by which surface winds anomaly induced by the ice sheets enhanced the AMOC. These studies should also be cited in this sentence.

> We thank the reviewer for pointing to a detailed list of very relevant references. We have cited them in the updated manuscript.

P13L31: Hu et al. (2015) investigated the impact of the closure of Bering Strait on the AMOC. This study should also be cited in this sentence.

> added; this reference is indeed very relevant.

5.2 MIS3 sensitivity to CO2 and ice sheet size
Results presented in this section are really interesting! As mentioned in Comment 1, I strongly encourage the authors to perform more analysis on these sensitivity experiments (surface salinity, density and sea ice cover over the NADW and AABW formation region, and the depth of the AMOC). Based on these analysis, you may further discuss the possible cause of differences among previous modelling studies. (Also, if possible, it may be interesting to discuss changes in surface air temperature and precipitation in the half-size Laurentide ice sheet. This analysis can provide an uncertainty of the simulated temperature and precipitation anomalies arising from the uncertainty in the shape of the MIS3 ice sheet. Just a suggestion.)

> Please see our response to the reviewer' general comment 1 regarding further analysis on the sensitivity experiments.

Discussion on the changes in temperature/precipitation in the modified ice sheet experiments would definitely be interesting to certain readership, as the reviewer suggested. Meanwhile, we would prefer to keep our focus on the AMOC bi-stability in this section and illustrate the relative insensitivity of NorESM MIS3 climate to external forcing. We would therefore not include this discussion in the manuscript, but rather leave it for future studies or for model intercomparison activities.

P15L13: What do you mean by 'ice inhibiting convection'?

> We mean "... Norwegian Sea are covered by sea ice that inhibits convection through its insulating effect."

Figures

Fig.2: Can you put labels on the contours?

> Yes, we have put labels on the contour lines 1000 m and 2000 m.

Fig.11: Can you add a figure showing the anomaly? It's difficult to understand the difference between MIS3 and PI from these figures.

> We tried to plot an anomaly map on the stream function, e.g. see the figure below. Comparing the two different ways of presenting, we think the original figure is relatively more straightforward and intuitive in comparing the stream functions during the two periods.

[Figure]

Figure above: MIS3 subtropical and subpolar gyre stream functions (Sv; contours) and difference with PI (Sv; shading)

References

Buizert, C., & Schmittner, A. (2015). Southern Ocean control of glacial AMOC stability and Dansgaard-Oeschger interstadial duration, Paleoceanography,  l30, doi:10.1002/2015PA002795

Hu AX, Meehl GA, Han WQ, Otto-Bliestner B, Abe-Ouchi A, Rosen-bloom N (2015) Effects of the Bering Strait closure on AMOC and global climate under different background climates. Prog Oceanogr 132:174–196. doi:10.1016/j.pocean.2014.02.004

Kawamura K et al (2017) State dependence of climatic instability over the past 720,000 years from Antarctic ice cores and climate modeling. Sci Adv 3:e1600446

Kobayashi H, Abe-Ouchi A, Oka A (2015) Role of Southern Ocean stratification in glacial atmospheric $CO_2$ reduction evaluated by a three-dimensional ocean general circulation model. Pale- oceanography 30:1202–1216. doi:10.1002/2015PA002786

Merkel U, Prange M, and Schultz M (2010) ENSO variability and teleconnections during glacial climates, Quat. Sci. Rev., 29, 86–100,.

Montoya M, Levermann A (2008) Surface wind-stress threshold for glacial Atlantic overturning. Geophys Res Lett 35:L03608. doi: 10.1029/2007gl032560

Oka A, Hasumi H, Abe-Ouchi A (2012) The thermal threshold of the Atlantic meridional overturning circulation and its control by wind stress forcing during glacial climate. Geophys Res Lett 39:L09709. doi:10.1029/2007gl029475

Sherriff-Tadano, S., Abe-Ouchi, A., Yoshimori, M., Oka, A., & Chan, W.-L. (2018). Influence of glacial ice sheets on the Atlantic meridional overturning circulation through surface wind change. Climate Dynamics, 50, 2881-2903. doi:10.1007/s00382-017- 3780-0

Sun, S., Eisenman, I., & Stewart A. (2016). The influence of Southern Ocean surface buoyancy forcing on glacial-interglacial changes in the global deep ocean stratification, Geophysical Research Letters, 43, 8124–8132. doi:10.1002/2016GL070058

---

## Author Comment (AC3) · 26 Feb 2019

Response to Reviewer #3:

We respond to the referee's comments in blue font below.

In the manuscript cp-2018-165 entitled "Equilibrium simulations of Marine Isotope Stage 3 climate" by Guo et al., the authors compared the simulated climate mean state of Marine Isotope Stage 3 (MIS3) and preindustrial (PI) era using the Norwegian Earth System Model (NorESM). They found a cooler climate in MIS3 relative to PI conditions with a thicker and more expanded sea ice. The AMOC strengthen by 13% with reduced AABW reaching the North Atlantic. Moreover the AABW production actually increases due to the increased sea ice cover in the southern oceans in association to the cooler MIS3 climate. They also show a reduced ENSO and NAM variability. Finally, by doing a few sensitivity simulations by reducing CO2 concentration or ice sheet height in the North America, they suggest that abrupt transitions of climate from interstadial to sta- dial state is not likely, and raised the question whether abrupt climate transition would be possible without changes of external forcings. I found this manuscript is well written and easy to follow. The results are interesting to the readership of the Climate Past community. Thus I would like to recommend this manuscript to be accepted after some revision:

We thank the reviewer for his/her overall positive comments on our manuscript. We respond to the reviewer's comments below point by point.

Comments:
1. The simulations are primarily focus on the PI and MIS3 climate background, thus it is not surprising that the climate states are stable for both conditions. One question the authors did not specifically clearly state is the initial condition of these runs. It seems that both PI and MIS3 runs state from the same ocean initial state, except an increase of the mean salinity for MIS3 run. Is this true? If so, how will this affect the model sensitivity when icesheet height or CO2 concentration changes?

> What the reviewer commented is true. We state in the manuscript that, for the MIS3 baseline experiment: "As for the PI experiment, the ocean model is initialised with modern temperature and salinity (steele et al., 2001) with the above mentioned salinity increment applied."; for the sensitivity experiments: "All the sensitivity experiments are branched off and initialised from the MIS3 interstadial simulation, and all other parameters are kept fixed."

The addition of extra salt to the global ocean in the MIS3 simulations must have some effects - although we expect them to be small - on the modelled ocean and climate background state, and therefore the model sensitivity to, e.g. ice sheet height and CO2 levels. As far as we are aware of, there has not been any study looking into this problem, which merits more investigations. This is highly relevant to any glacial simulations, especially to LGM experiments where PMIP protocols define a global salinity addition of +1 psu.

2. The authors tested the model response of CO2 reduction be 15 ppmv. The question here is whether the MIS3 stadial climate is caused by CO2 reduction or by changes of the AMOC? It seems that the authors assumed that the CO2 reduction is the cause, is it true?

> We did not assume that the CO2 reduction is the cause of the stadial climate state. We support the wide-accepted view that changes in AMOC (e.g. strong or weak/off mode) are behind the interstadial/stadial climate states. As described in the Introduction, changes in AMOC can be invoked by changes in CO2 concentrations, as well as other factors such as changes in the size of Laurentide Ice Sheet.

The reason why we do such a sensitivity experiment (15 ppmv lower CO2), as also discussed later in the text, is partly motivated by some previous studies (e.g. Zhang et al. 2017; Klockmann et al. 2018) that do indeed show a transition of AMOC mode upon relatively small change of CO2 level. Another motivation is that if the model is already close to the threshold of mode change, which we did not know beforehand, a small

change of external forcing is able to kick the system into a different state (see also our response to the comment below).

3. Although the experiments done by the authors don't show significant AMOC changes, it seems this is not enough to question the possible multi-equilibria of AMOC, especially the experimental design may not serve the purpose of the authors. A better test is to check whether AMOC has multi-equilibria in the NorESM under glacial condition. If yes, then the authors can test whether an abrupt transition of the AMOC is possible with the absence of the external forcing change. It may be important to test the small changes of the external forcings and whether this small changes can bring the climate state to a critical point in which even smaller changes in freshwater forcing is capable to collapse the AMOC.

> We agree with the reviewer that given the experiments performed, we cannot question the multi-equilibria of the AMOC; rather, the experiments suggest that our simulated MIS3 climate stays far away from the bifurcation/tipping point, and is in contrast to some previous studies that show 'sweet spot' within a certain range of external forcing, therefore addressing model dependence in studying model bi-stability.

We also agree that a more thorough and systematic design of model experiments are needed in exploring the multi-equilibria and hysteresis behaviour of the model, forced with a wider range of external forcing. Once we are close to the 'threshold', a small change of forcing is expected to be able to tip the system from one state to another - or even with self-sustained climate transitions. We realise that such studies are certainly meaningful, but are beyond the scope of the current study, and are worth further investigations in the future.

---

## Author Comment (AC4) · 26 Feb 2019

Response to Reviewer #4:

We respond to the referee's comments in blue font below.

The manuscript "Equilibrium simulations of Marine Isotope Stage 3 Climate" by Guo and colleagues well present a new MIS3 simulation in their model NorESM1-F. They employed a latest (new) ice sheet configuration without a large Fennoscandian ice sheet to conduct their MIS3-38ka simulation. By several attempts exploring the tipping point/bifurcation in their 38ka simulation, the authors claim that their 38ka simulation in NorESM1-F is too stable to reach a tipping point that was commonly used to explain the millennial-scale variability during the MIS3 in previous modeling studies.

We thank the reviewer for his/her constructive comments on our manuscript. We respond to the reviewer's comments below point by point.

The authors first described their 38ka simulation results in very detail (although it can be more compact), in accompany with a comparison with previous LGM simulations. The LGM is a good reference (LGM) to compare with, but there is lack of detailed discussion of their differences. The 38ka simulation was integrated for 2500 model years. The author argue that their simulation is almost in a quasi-equilibrium since the salinity trend in the Atlantic is less than 0.06 g/kg. How is it defined because it remains possible that the deep ocean is not in a quasi-equilibrium state if there exists a robust salinity increase in the AABW formation region. This is also my concern for their sensitivity simulation with lower $CO_2$ levels. It is very likely that polar regions need a much longer time scale (> 1000 model years) to cool down, producing the cold enough bottom/deep water masses and so the glacial ocean structure and circulation.

> Following the reviewer (and another reviewer)'s comment on the detailed description of the simulation results, we have significantly

reduced the length of the manuscript by moving some texts/figures/sections to the supplementary material; these include the time series of sea ice, several atmospheric diagnosis, the whole section of "Modes of variability", and part of the "stadial" experiment. The updated paper structure reads more compact than the previous version.

Regarding the reviewer's comment on comparison with previous LGM simulations, we actually tried to avoid a detailed discussion, as the two periods are distinct with each other in certain important aspects, e.g. MIS3 features reduced ice volume and a relatively warm climate compared to the LGM. In the updated manuscript, we've added more discussions with the existing MIS3 studies which were not adequately addressed in the previous version.

As for the length of model simulation, we are aware that the deep ocean, in particular the deep Pacific, cannot be fully ventilated. However, the ideal age in the deep Pacific by the end of model integration (2500 years) is less than 1600 years, indicating a relatively well ventilated ocean. There is not any 'hard' threshold to define the model state of quasi-equilibrium in the paleoclimate simulations. The only proposed threshold for the glacial simulations was proposed by Zhang et al. (2013), e.g. a salinity trend < 0.006 g/kg per century in the deep Atlantic can be considered as quasi-equilibrium. As for the lower $CO_2$ experiments, one integrated for 800 years and the other 1200 years, we agree that both runs can be extended. The experiments, with their current length of integration, do suggest that our simulated MIS3 climate stays far away from the bifurcation/tipping point, and is in contrast with previous studies that show 'sweet spot' within a certain range of external forcing, therefore addressing model dependence in studying model bi-stability.

The authors also assess the simulated climate mode variability in their simulations. It makes the manuscript more comprehensive. However, I do not find a clear connection to the following investigation of the AMOC bistability? why does the "stadial" condition under 40ka boundary not include freshwater forcing in the North Atlantic? I fully understand the

authors' purpose, but since H4 did feature a robust freshwater input, it would be more comparable and reasonable to force a stadial climate with the North Atlantic freshwater forcing under 40ka-38ka boundary conditions. If the authors would like to investigate the changes in climate variability in stadial conditions, I would suggest conducting the hosing under 38ka boundary condition. This will largely reduce the difficulty in the discussion of differences between interstadial and stadial runs. The present "stadial" climate can be included in the section regarding exploration of AMOC bistability.

> Following the reviewer' comment, we have moved the section on climate variability and the majority of the "stadial" experiment to the supplementary material, and the rest of the discussion on the "stadial" experiment to the AMOC bi-stability section.

We set up the "stadial" experiment to test if a non-Henrich stadial can be simulated with NorESM, as motivated by some previous studies (e.g. Zhang et al. 2017; Klockmann et al. 2018) that show a transition of AMOC mode and therefore a stadial-like climate upon relatively small change of $CO_2$ level. As the reviewer pointed out, freshwater flux is indeed present at H4, though with large uncertainties in magnitude and location, and should be applied for a 'realistic' stadial climate simulation. This is our ongoing work that we intend to write up in a follow up study; we use the hosing experiments with 38 ka BP boundary conditions to study the transition process from H4 to GI8.

In the discussion part, the author design several sensitivity experiments to explore the potential nonlinear behavior of their 38ka climate. The experiments are reasonable and clear, which can provide end-members of climate responses to glacial-interglacial variations regarding ice volume and $pCO_2$. It is a promising try here although probably these runs (especially lower $CO_2$ runs) are not in quasi-equilibrium, therefore the runs are not as conclusive as the authors argued. In addition, one conceptual mistake in the manuscript is that spontaneous oscillation does not share the same definition with the AMOC bistability, but rather a hopf bifurcation feature. The authors shall go through the manuscript carefully to distinguish their difference.

Overall I find the manuscript is interesting and well within the scope of Clim. Past. It can be accepted for publication after some modifications in the structure as well as refinements of model results description and discussion.

> We thank the reviewer for the positive comments on our manuscript. As we responded above, we agree that extending the sensitivity experiments, together with a wider forcing range, could be more beneficial to the simulation results and conclusions.

We went through the manuscript to double check the conceptual issue pointed out by the reviewer (e.g. see the second and third paragraphs of Section 4.3).

Finally, following all the reviewers' comment, we've made efforts to restructure the manuscript and present the results in a more succinct way.

---

## Referee Report (RR1)

**Equilibrium simulations of Marine Isotope Stage 3 climate by Guo et al. - second review:**

In the revised manuscript, the authors made progress in reducing the amount of the manuscript and in clarifying the significance of the study. New analysis and discussions on the sensitivity experiments are presented, showing changes in SSS, sea ice, and the depth of the AMOC in response to lowering of CO2 and the flattening of the Laurentide ice sheet. The anomalies in SSS and sea ice are small compared with other climate model, and emphasises the importance of model comparison. In total, I'm satisfied with the revised manuscript, and recommend this paper for publication after several specific comments are revised.

**Specific comments**

**1 Introduction**

P2L4: "illusive" -> "elusive"?

P2L28: I found another MIS3 modelling study, which conducted a simulation of 32ka with COSMOS AOGCM (Gong et al. 2013). You may cite this paper as well.

**2 Method**

P4L29: I noticed that the Laurentide ice sheet is merged with the Cordilleran ice sheet. While the shape of the MIS3 ice sheet is highly uncertain, other studies also show separated Laurentide and Cordilleran ice sheets before LGM (e.g. Abe-Ouchi et al. 2007). Furthermore, some studies shows a significant impact of merging of these two ice sheets on the climate (e.g. Lofverstrom et al. 2014). Given these background, it is worth mentioning the uncertainty in the reconstruction of ice sheet as well.

**3 Result**

P7L1: Why don't you remove "leading to more open ocean convection" from this sentence? Indeed, the increase in sea ice plays an important role in increasing the formation of AABW, but it doesn't have to be via open ocean convection.

P8L5: It would be better to say "The reduced level of CO2 during the MIS3 causes a lowering of SST and an expansion of sea ice." , as changes in ice sheet can have a significant impact on the SST at some regions (e.g. Northwestern Pacific e.g. Fig. S13e).

P8L19: It would be better to say "The cooling in the North Pacific is *partly* associated with ~" since the cooling is also induced by the lowering of CO2.

**4 Discussion**

P11L23: I'm not sure about this sentence. As the authors describe, some studies suggest a vigorous AMOC during the LGM, however there are also several other studies suggesting a weak AMOC during the LGM (e.g. McManus et al. 2004). These studies should also be referred.

P11L28: "at *the* LGM"

P14L5: probably "160 ppmv" rather than "180 ppmv"

P14L15:Thank you for adding these figures. They are very useful in comparing the results with other models and also should increase the impact of this paper. Indeed, the changes in sea surface salinity (SSS) are small compared with other climate model studies (e.g. Fig. 3d in Klockmann et al. 2016). These results imply the importance of model comparison.

P17L11: probably "Fig. S11" -> "Fig. S13"

**Figures**

Fig. S4: I agree that the anomaly field is distorted by the difference in the land-sea mask between MIS3 and PI. However, Fig. S4 is still difficult to interpret. It would be better to show the results of MIS3 and PI in different figures. In addition, putting the labels on the contour may help the readers to identify the difference between MIS3 and PI.

Fig. S12-14: Please describe the period of the sensitivity experiments used to calculate the anomaly figures.

Abe-Ouchi A, Segawa T, Saito F (2007) Climatic Conditions for modelling the Northern Hemisphere ice sheets throughout the ice age cycle. Climate of the Past, 3, 423-438, doi: 10.5194/cp-3-423-2007.

Gong, X., Knorr, G., Lohmann, G., and Zhang, X. (2013) Dependence of abrupt Atlantic meridional ocean circulation changes on climate background states, Geophys. Res. Lett., 40, 3698–3704, doi:10.1002/grl.50701

Löfverström M, Caballero R, Nilsson J, Kleman J (2014) Evolution of the large-scale atmospheric circulation in response to changing ice sheets over the last glacial cycle. Climate of the Past, 10, 1453-1471, doi:10.5194/cp-10-1453-2014.

McManus JF, Francois R, Gherardi JM, Keigwin LD, Brown-Leger S (2004) Collapse and rapid resumption of Atlantic meridional circulation linked to deglacial climate changes. Nature, 428, 834-837, doi:10.1038/nature02494.

---

## Author Response (AR2)

Response to Reviewer #1:

Please see response to the referee's comments in blue font.

In the revised manuscript, the authors made progress in reducing the amount of the manuscript and in clarifying the significance of the study. New analysis and discussions on the sensitivity experiments are presented, showing changes in SSS, sea ice, and the depth of the AMOC in response to lowering of CO2 and the flattening of the Laurentide ice sheet. The anomalies in SSS and sea ice are small compared with other climate model, and emphasises the importance of model comparison. In total, I'm satisfied with the revised manuscript, and recommend this paper for publication after several specific comments are revised.

We are glad that the reviewer is satisfied with our revision of the manuscript. We respond to the reviewer's comments below point by point.

Specific comments

1 Introduction
P2L4: "illusive" -> "elusive"?

> yes, indeed!

P2L28: I found another MIS3 modelling study, which conducted a simulation of 32ka with COSMOS AOGCM (Gong et al. 2013). You may cite this paper as well.

> Thanks for suggesting the reference. We have cited this paper in the updated manuscript.

2 Method
P4L29: I noticed that the Laurentide ice sheet is merged with the Cordilleran ice sheet. While the shape of the MIS3 ice sheet is highly uncertain, other studies also show separated Laurentide and Cordilleran ice sheets before LGM (e.g. Abe-Ouchi et al. 2007). Furthermore, some studies shows a significant impact of merging of these two ice sheets on the climate (e.g. Lofverstrom et al. 2014). Given these background, it is worth mentioning the uncertainty in the reconstruction of ice sheet as well.

> We acknowledge that large uncertainties exist regarding the reconstructions of the ice sheets during MIS3. Following the reviewer's comment, we have added the following to the manuscript – "The Laurentide Ice Sheet is merged with the Cordilleran Ice Sheet in our MIS3 ice sheet configuration. However, the shape of the MIS3 ice sheet is highly uncertain; studies have also shown separated Laurentide and Cordilleran Ice Sheets before the LGM (e.g. Abe-Ouchi et al. 2007)."

We think that it is not immediately clear regarding the impact of the merging/separation of the Laurentide and Cordilleran Ice Sheets on the Northern Hemisphere climate, e.g. in the studies by Löfverström et al. (2014), the effects of the separation of the two ice sheets (their Fig. 2c) is masked by the effects of a much larger Laurentide Ice Sheet in the merged case (their Fig. 2d). We are not aware of any studies on a "clean" comparison on the effects of merging, or separation of the two ice sheets (e.g. with a Laurentide Ice Sheet of the same size).

3 Result
P7L1: Why don't you remove "leading to more open ocean convection" from this sentence? Indeed, the increase in sea ice plays an important role in increasing the formation of AABW, but it doesn't have to be via open ocean convection.

> Following the reviewer's suggestion, we have removed "leading to more open ocean convection" from the sentence.

P8L5: It would be better to say "The reduced level of $CO_2$ during the MIS3 causes a lowering of SST and an expansion of sea ice." , as changes in ice sheet can have a significant impact on the SST at some regions (e.g. Northwestern Pacific e.g. Fig. S13e).

> modified.
P8L19: It would be better to say "The cooling in the North Pacific is partly associated with ~" since the cooling is also induced by the lowering of $CO_2$.

> modified.

4 Discussion
P11L23: I'm not sure about this sentence. As the authors describe, some studies suggest a vigorous AMOC during the LGM, however there are also several other studies suggesting a weak AMOC during the LGM (e.g. McManus et al. 2004). These studies should also be referred.

> Reconstructions of AMOC during glacial times are a big topic and cannot be adequately summarized within a sentence or two (see the review paper by Lynch-Stieglitz, 2017). We think that the current available reconstructions may describe different scenarios, but broadly point to a shallower AMOC of similar strength compared to today. In particular, combining the Pa/Th with the ratio of neodymium isotopes (εNd), as in Böhm et al. (2015), could produce a more robust reconstruction of the glacial state of the AMOC. The combined measurements, as also reported by other recent studies (e.g. Lippold et al., 2016), point to a strong yet shallow upper cell of the AMOC at the LGM. To keep our discussion focused, we prefer not to expand our discussion and review the myriad reconstructions of the AMOC.

Lynch-Stieglitz, The Atlantic Overturning Circulation and Abrupt Climate Change, Annual Review of Marine Science 9, 83-104, 2017.
Lippold et al., Deep water provenance and dynamics of the (de)glacial Atlantic meridional overturning circulation, Earth and Planetary Science Letters, 445, 68–78, 2016.

P11L28: "at the LGM"

> modified.

P14L5: probably "160 ppmv" rather than "180 ppmv"

> modified.

P14L15:Thank you for adding these figures. They are very useful in comparing the results with other models and also should increase the impact of this paper. Indeed, the changes in

sea surface salinity (SSS) are small compared with other climate model studies (e.g. Fig. 3d in Klockmann et al. 2016). These results imply the importance of model comparison.

> We thank the reviewer for the suggestion on adding these results. We agree with the reviewer that model intercomparison is important and necessary in simulating abrupt climate changes.

P17L11: probably "Fig. S11" -> "Fig. S13"

> (P14L11 not P17L11) modified.

Figures
Fig. S4: I agree that the anomaly field is distorted by the difference in the land-sea mask between MIS3 and PI. However, Fig. S4 is still difficult to interpret. It would be better to show the results of MIS3 and PI in different figures. In addition, putting the labels on the contour may help the readers to identify the difference between MIS3 and PI.

> The anomaly field is not much distorted by the difference of the land-sea mask, rather, the difference between the two states is not immediately clear, considering that the subpolar gyre during MIS3 is not dramatically stronger than (e.g. broadly similar with) that in the PI.

Plotting the subpolar gyres separately would not make it easier to show the difference between the two (as the streamlines are located in similar places between the two experiments in the majority of the North Atlantic). To aid interpretation, we have added the following in the caption – "The barotropic subtropical and subpolar gyres are broadly similar for the MIS3 and PI experiments; additional streamlines in the North Atlantic subpolar region during MIS3 indicate a stronger subpolar gyre."

Fig. S12-14: Please describe the period of the sensitivity experiments used to calculate the anomaly figures.

> The anomalies are the average over the last 100 years of each experiment. We have added the following information in the captions – "The MIS3 control state is the average from years 1901-2000, and the anomalies are averages over the last 100 years of each experiment."

Last but not least, we appreciate the reviewer's great efforts, including a thorough assessment and constructive comments throughout the review process. This has significantly helped improve the manuscript!

Abe-Ouchi A, Segawa T, Saito F (2007) Climatic Conditions for modelling the Northern Hemisphere ice sheets throughout the ice age cycle. Climate of the Past, 3, 423-438, doi: 10.5194/cp-3-423-2007.

Gong, X., Knorr, G., Lohmann, G., and Zhang, X. (2013) Dependence of abrupt Atlantic meridional ocean circulation changes on climate background states, Geophys. Res. Lett., 40, 3698–3704, doi:10.1002/grl.50701

Löfverström M, Caballero R, Nilsson J, Kleman J (2014) Evolution of the large-scale atmospheric circulation in response to changing ice sheets over the last glacial cycle. Climate of the Past, 10, 1453-1471, doi:10.5194/cp-10-1453-2014.

McManus JF, Francois R, Gherardi JM, Keigwin LD, Brown-Leger S (2004) Collapse and rapid resumption of Atlantic meridional circulation linked to deglacial climate changes. Nature, 428, 834-837, doi:10.1038/nature02494.

The manuscript by Guo and colleagues has greatly improved after the first round of reviews. The new insights provided by the study and its place in the context of previous MIS3 studies comes out much more clearly. The discussion of the AMOC response has also become more clear.
I recommend to accept the manuscript subject to a few minor revisions which I list below:

We are glad that the reviewer is satisfied with our revision of the manuscript. We respond to the reviewer's comments below point by point.

1. The PI simulation should also be introduced in Section 2.2. If the PI simulation is the same as the simulation in the Guo et al (2019, GMD) paper, then this should at least be mentioned - either in section 2.1 or towards the end of section 2.2, when the PI simulation is first referred to.

> Indeed. We have added the following at the end of section 2.2 – "The PI experiment, that the MIS3 experiment is evaluated against, is performed with the same model, NorESM1-F, and is documented and assessed in Guo et al., (2019)."

2. (p.7, ll-14-18) It would be nice to have some conclusion from the comparison of LGM and MIS3 surface temperatures. Is the result of the comparison reasonable given the differences and similarities between MIS3 and LGM?

> Agreed. We have added the following in the manuscript – "The simulated smaller cooling during MIS3, relative to the LGM, is associated with lower $CO_2$ and a smaller ice sheet."

3. (p.6, ll.30-31) When describing the spin-up of the AMOC in section 3.1, you say that the final difference in AMOC strength is 1.9Sv. But this difference at 26N appears to be an underestimation of the total difference, as the difference in max. AMOC strength seems to be 3.2 ( p.9, l.17). This should at least be mentioned.

> We agree that it is useful to list both numbers of difference at the same place. Following the reviewer's suggestion, we have added the following to the manuscript – "Similarly, the simulated maximum strength of the AMOC is 27.5 Sv, 3.2 Sv stronger than that in the PI (24.3 Sv)."

4. Two sections have been moved completely into the supplementary material (precipitation on p.8 and modes of variability on p.11). This is very beneficial for the overall length and readability of the manuscript. But I would suggest that at least one or two sentences should be included in the manuscript to summarise the main findings of the sections that have been moved. To have only the reference to the existence of the extra sections without connecting them to main text seems a bit odd. It raises the question whether they are needed at all.

> This is an excellent comment.

Following the reviewer's suggestion, we have added the following to the main manuscript text (page 8):
"In the colder MIS3 climate, simulated global mean precipitation is 0.18 mm day$^{-1}$ lower, with both seasonal and geographical changes including a southward shift of the Intertropical Convergence Zone (ITCZ) compared to that in PI. More details are given in the Supplementary Section 2."

and (page 11):
"Simulated ENSO variability is weakened in the MIS3 compared to the PI simulation. For the NAM, simulated centres of action over the Arctic and North Pacific are both weakened, with the latter much reduced due to the presence of the elevated Laurentide Ice Sheet. More details are given in the Supplementary Section 3."

5. (p.10, ll.24-25) It is not immediately clear to me, how the stronger northerly winds lead to less sea ice in the western Labrador Sea.

> We expect the simulated strong northerly wind anomaly during MIS3 to help move sea ice away from the area. We have rephrased the sentence as – "However, there is less sea ice in the western Labrador Sea, which is likely due to the strong northerly katabatic wind induced by the presence of the adjacent Laurentide Ice Sheet (Fig. 6) that moves the ice away."

6. (p.11, ll.27-29) An additional factor could be the active deep water formation in the Nordic Seas in the MIS3 simulation. Under LGM conditions I would expect the Nordic Seas to be completely ice covered with little deep convection there. This could also contribute to the different AMOC state.

> This is also a reasonable and plausible explanation. However, we are not aware of any relevant references and have not included this point.

7. (p.14, ll.23-24) Add a reference to Figure S6 here.

> added.

I look forward to seeing the manuscript in its final and published form!

Last but not least, we appreciate the reviewer's great efforts in giving a thorough assessment and constructive comments throughout the review process. This has helped significantly in improving the manuscript.

[revised manuscript text omitted]